# 3D directional tuning in the orofacial sensorimotor cortex during natural feeding and drinking

Victoria B Hosack[1]*, Fritzie Arce-McShane[1,2,3]*

[1]Department of Oral Health Sciences, School of Dentistry, University of Washington, Seattle, United States; [2]Division of Neuroscience, Washington National Primate Research Center, University of Washington, Seattle, United States; [3]Graduate Program in Neuroscience, University of Washington, Seattle, United States

## eLife Assessment

This study characterises motor and somatosensory cortex neural activity during naturalistic eating and drinking tongue movement in nonhuman primates. The data, which include electrophysiology, three-dimensional tracking of tongue movements, and nerve block manipulations, are **valuable** to neuroscientists and neural engineers interested in tongue use. Although the current analyses provide a **solid** description of single neuron activity in these areas, both the population level analyses and the characterisation of activity changes following nerve block could be improved.

*For correspondence:
vhosack@uw.edu (VBH);
fritziea@uw.edu (FA-McS)

Competing interest: The authors declare that no competing interests exist.

**Abstract** Directional tongue movements are crucial for feeding and speech, ensuring proper food positioning for chewing and swallowing, as well as accurate sound production. While directional tuning in the arm region of the sensorimotor cortex during reaching tasks is well studied, little is known about how three-dimensional (3D) tongue direction is encoded in the orofacial sensorimotor cortex (OSMCx) during natural behaviors. Understanding this neural representation has important implications for rehabilitating individuals with orolingual dysfunctions. This study examines the directional tuning and population dynamics in OSMCx during naturalistic feeding and drinking, and how these are affected by sensory loss. Using biplanar video-radiography, we tracked implanted tongue markers in behaving rhesus macaques (*Macaca mulatta*) and simultaneously recorded 3D positional data with spiking activity from chronically implanted microelectrode arrays in primary motor (MIo) and somatosensory (SIo) areas of the orofacial cortex. In some sessions, tasks were preceded by bilateral nerve block injections to the sensory branches of the trigeminal nerve. Modulation to 3D tongue direction during feeding and drinking was found in most MIo and SIo neurons. Directional information at both individual and population levels was higher in feeding and was more robust in MIo. Following sensory loss, alterations in tongue kinematics were accompanied by changes in directional information in MIo and SIo, manifesting as modifications in both individual neuron tuning characteristics and the broader dynamics of population-level neural activity. This study advances our understanding of single-neuron and population activity in OSMCx and their potential contributions to the sensorimotor control of complex naturalistic tongue movements. By extending current knowledge of orofacial control to 3D tongue movements, our findings demonstrate the specificity and adaptability of population activity in MIo and SIo in response to different behavioral contexts, providing important insights for understanding neural mechanisms underlying skilled tongue control.

## Introduction

Motor and somatosensory cortical neurons modulate their spiking activity based on movement direction as seen in arm reaching tasks (*Georgopoulos et al., 1988*; *Schwartz et al., 1988*; *Prud'homme and Kalaska, 1994*) and orofacial behaviors. In the primary motor (MIo) and primary somatosensory (SIo) areas of the orofacial sensorimotor cortex (OSMCx), neurons encode the direction of voluntary tongue protrusion (*Murray and Sessle, 1992*; *Lin et al., 1994*) and semiautomatic tongue movements in chewing and swallowing (*Sessle et al., 2005*). Extensive research has explored how the arm region of the sensorimotor cortex encodes movement direction (*Ajemian et al., 2000*; *Georgopoulos et al., 2007*; *Churchland et al., 2012*; *Lillicrap and Scott, 2013*). Since the tongue is enclosed within the oral cavity and thus hidden from view, it has proved difficult to study the neuromechanical processes underlying directional tongue movements that are essential for these behaviors (*Hiiemae and Palmer, 2003*). Thus, considerably less is known about how 3D tongue direction is encoded in the OSMCx and the role of tactile sensation (*Bach-y-Rita et al., 1998*; *Lamm et al., 2005*; *Lozano et al., 2009*) during natural feeding and drinking. This knowledge has important implications for improving evaluation and treatment strategies for individuals with sensorimotor dysfunctions (*Takizawa et al., 2016*; *Avivi-Arber and Sessle, 2018*).

The OSMCx plays an important role in the coordination of complex tongue movements. Seminal studies on the directional tuning properties of OSMCx neurons by Sessle and colleagues employed varying locations of spouts that delivered a juice reward to elicit directional tongue protrusions without tracking tongue movements. A later study incorporated tracking of 2D tongue movements using videofluoroscopy during voluntary directional protrusions (*Arce et al., 2013*), but the tongue trajectories were not used to study directional tuning. In all these prior studies, primates have been trained to interact with a computer display to elicit a tongue protrusion to a specific direction on cue. There is a knowledge gap on how spiking activity in the OSMCx relates to tongue movements during natural behaviors. With the development of biplanar video-radiography (*Brainerd et al., 2010*), it is now possible to track these 3D tongue movements within the oral cavity at a high temporal and spatial resolution (*Montuelle et al., 2020*; *Feilich et al., 2021*). By simultaneous recording of precise tongue movements and spiking activity, we have shown recently that tongue position and shape can be accurately decoded from OSMCx during feeding (*Laurence-Chasen et al., 2023*).

Our study investigated how OSMCx encodes and decodes tongue direction during untrained feeding and drinking, comparing activity across multiple cortical regions. Given the importance of oral somatosensation in tongue positioning and bolus control during chewing and swallowing (*Smith*

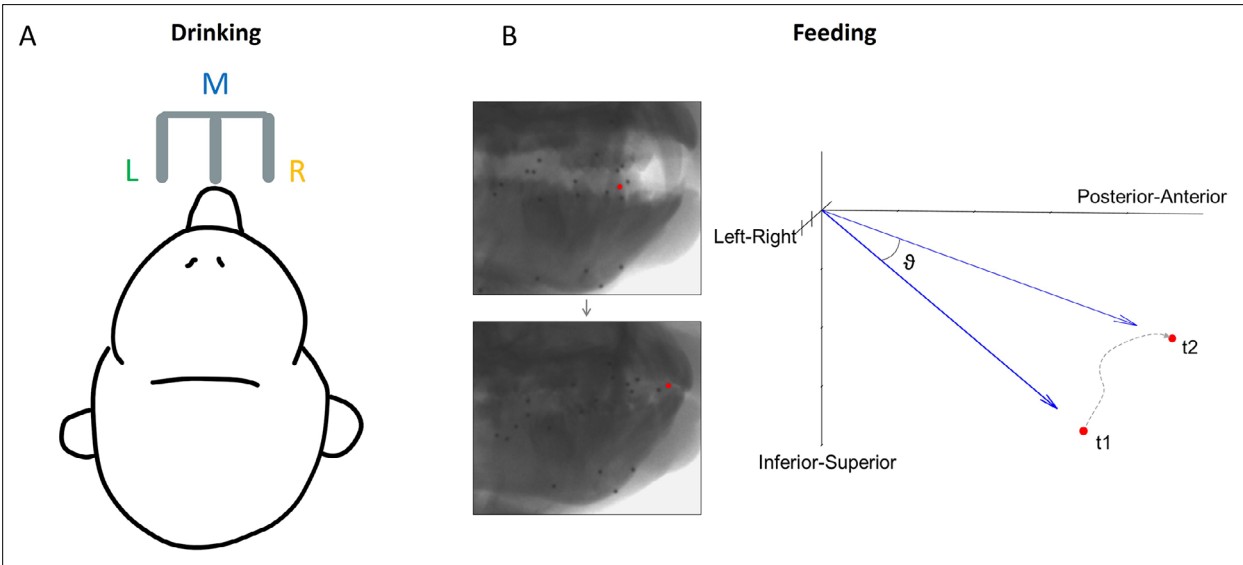

**Figure 1.** Direction of tongue motion in each behavioral task. (**A**) Schematic of the location of three spouts, left (L), middle (M), and right (R), for the drinking task. Tongue direction was categorized based on spout location. (**B**) Calculation of 3D tongue direction during feeding. $\theta$ is the instantaneous 3D direction of the tongue tip over a 100 ms interval between its positions at t1 and t2, where t1=0 and t2=t1+100. The dotted line shows the actual trajectory during this interval.

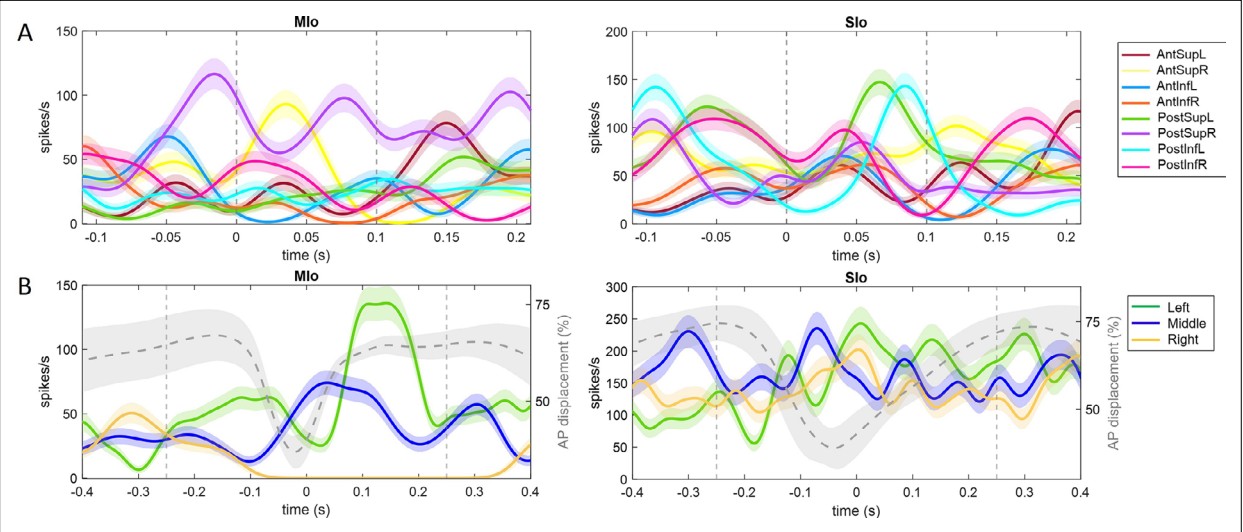

**Figure 2.** Examples of single neuron activity in relation to tongue direction. (**A**) Each peri-event time histogram (PETH and ±1 SE, smoothed by a 25 ms Gaussian kernel) corresponds to spiking activity for a specific range of tongue direction for feeding trials. Dashed lines indicate 100 ms interval used for calculating the tongue direction. (**B**) PETHs for drinking trials with the same spout, centered at the point of minimum protrusion of the tongue (0 s). Percent tongue displacement along the anterior-posterior axis is shown in gray, with shaded area representing ±1 SD. Vertical lines indicate 500 ms interval used for tuning analysis.

The online version of this article includes the following figure supplement(s) for figure 2:

**Figure supplement 1.** Directional neural responses across trials.

**Figure supplement 2.** Microelectrode array locations.

_and Cutrer, 2011_), we also examined its role in sensorimotor control by selectively blocking oral tactile sensation.

## Results

Previous research has investigated directional tuning of OSMCx neurons during trained tongue protrusion tasks. Our study extends this work by investigating two natural, untrained behaviors: feeding and licking ('drinking') from a spout (_Figure 1_). These behaviors provide distinct contexts for studying tongue coordination - feeding allows unrestricted tongue movement, while drinking constrains movement direction based on spout location. This comparison allows us to analyze how OSMCx's encoding of the 3D direction of tongue movements varies across behavioral contexts.

### Neuronal modulation patterns differ between MIo and SIo

Many neurons exhibited significant modulation of spiking activity to tongue direction (bootstrap, $p < 0.05$), though there were diverse patterns. _Figure 2_ shows peri-event time histograms (PETHs) of two example neurons related to tongue movements during feeding and drinking. In feeding, both neurons showed complex oscillatory firing, with notable peaks between –0.05 s and 0.1 s for upward movements to the right (MIo, _Figure 2A_, left) and posterior movements to the left (SIo, _Figure 2A_, right). For drinking, there was clear separation between different directions in the MIo neuron, with the left (green) exhibiting the highest activity (_Figure 2B_, left), while SIo showed oscillatory patterns with less distinct separation between spout locations, but higher overall activity (_Figure 2B_, right). Like the arm region, the tuning curves of directionally modulated MIo and SIo neurons fit the cosine tuning function (F-test, $p < 0.05$, feeding: MIo = 86%, SIo = 75%). _Figure 3A_ maps a neuron's firing rate for tongue movements in left-right, inferior-superior, and posterior-anterior axes. Here, an MIo neuron is strongly tuned to posterior-anterior and inferior-superior directions, while remaining unresponsive to left-right movements during natural feeding. Many of the recorded neurons in each population behaved in a similar fashion, with peaks most frequently observed toward the anterior and superior directions. This observation was consistent with the tongue movements being most frequent in the

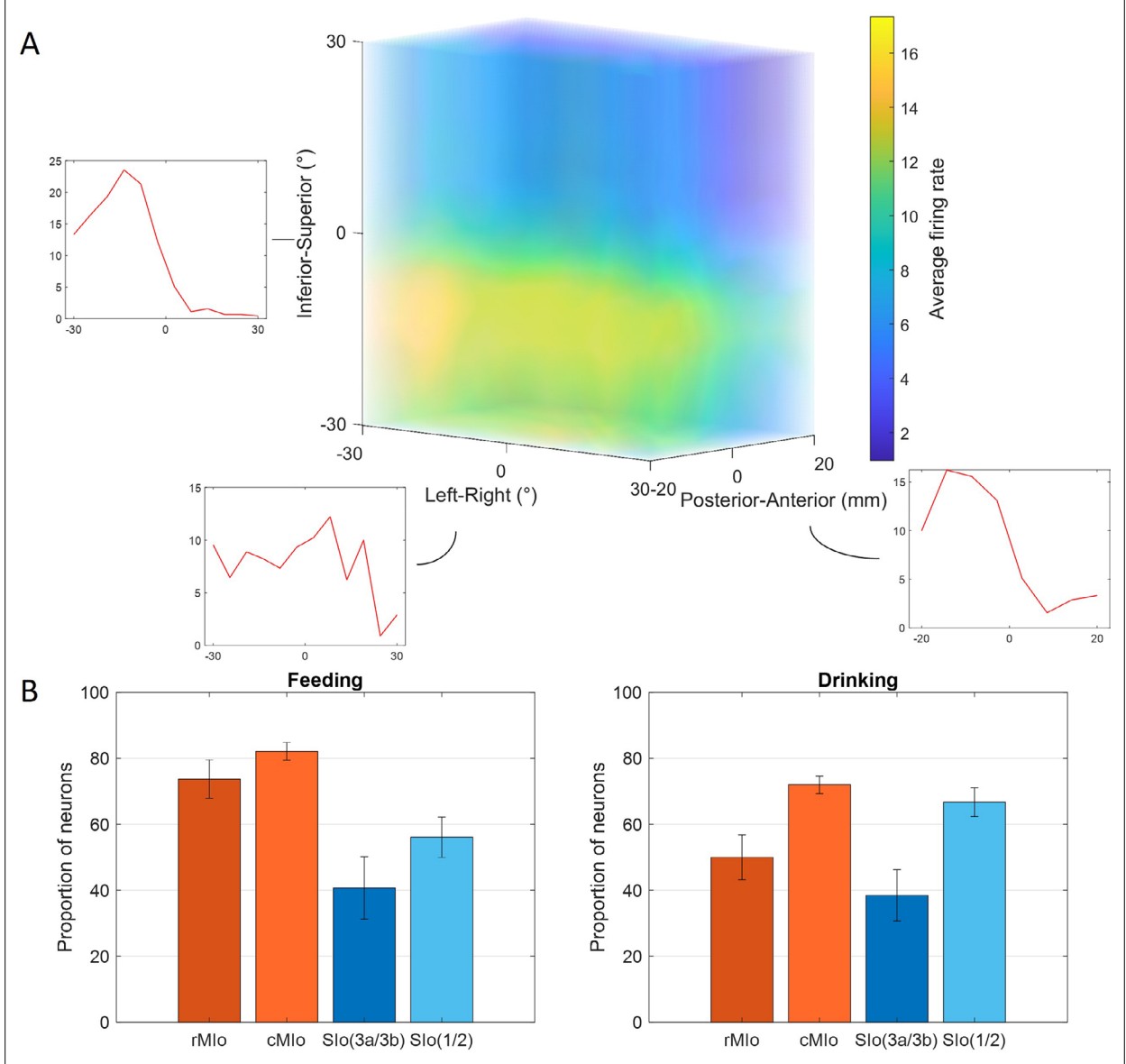

**Figure 3.** Directional tuning of neurons during control tasks. (**A**) 3D firing rate map of a neuron in MIo during feeding. Smaller inset plots are 1D tuning curves across each axis. (**B**) Percentage of neurons tuned to direction, combined for both subjects. Recordings were taken from four areas of the orofacial sensorimotor cortex (OSMCx): rMIo - rostral M1 (n = 57, 54), cMIo - caudal M1 (n = 201, 289), SIo(3a/3b) - area 3a/3b (n = 27, 39), and SIo(1/2) - area 1/2 (n = 66, 117). Error bars represent ±1 SE.

The online version of this article includes the following source data and figure supplement(s) for figure 3:

**Source data 1.** *Figure 3B*: Percentage of directionally tuned neurons.

**Figure supplement 1—source data 1.** Number of feeding trials in each group of directions.

**Figure supplement 1.** Proportion of feeding trials in each group of directions.

**Figure supplement 2.** Proportion of neurons directionally modulated during chews vs. swallows in both monkeys.

**Figure supplement 2—source data 1.** Directionally modulated neurons during chews versus swallows.

anterior-superior directions, followed by the posterior-inferior (*Figure 3—figure supplement 1*). The varying neuronal responses to tongue direction could not be attributed to variability in their firing, as the distribution of the Fano factor was similar across directions (Kruskal-Wallis, $p>0.1$ for all except SIo control drinking, $p=0.06$).

The proportion of directionally tuned neurons was higher in the feeding vs. drinking task (chi-square, $p<0.05$, feeding: 72%, drinking: 66%) and differed significantly between MIo and SIo during

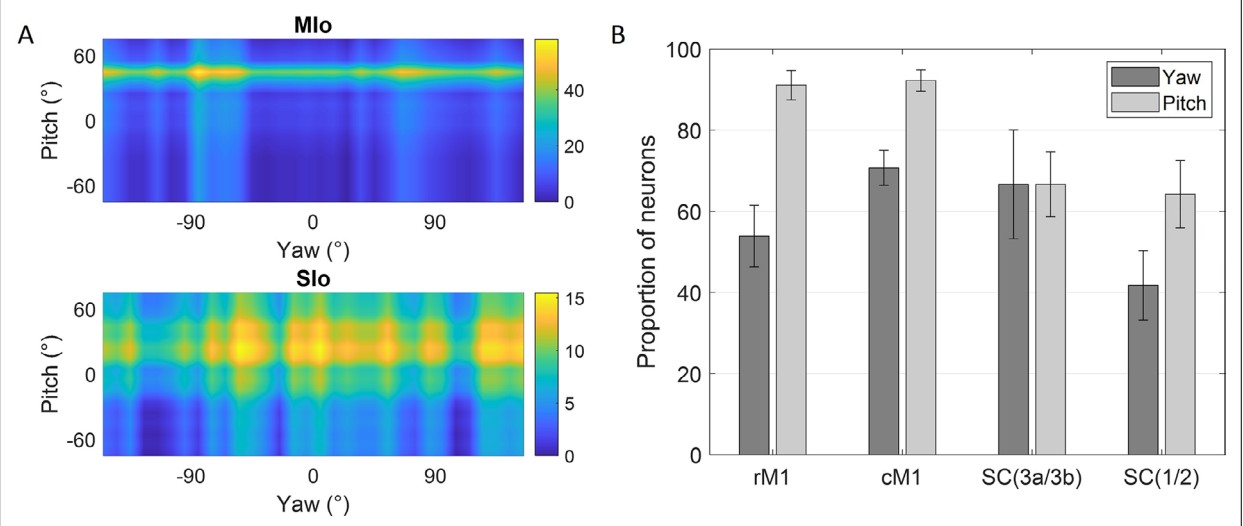

**Figure 4.** Directional tuning to yaw and pitch during feeding. (**A**) Firing rate maps of a neuron in MIo and in SIo across yaw and pitch angles. Firing rates were averaged across all 100 ms feeding intervals within a 10° range. (**B**) Proportion of neurons tuned to yaw and pitch, combined for both subjects. Recordings were taken from four areas of the orofacial sensorimotor cortex (OSMCx): rMIo - rostral M1 (n = 57, 54), cMIo - caudal M1 (n = 201, 289), SIo(3a/3b) - area 3a/3b (n = 27, 39), and SIo(1/2) - area 1/2 (n = 66, 117). Error bars represent ±1 SE.

The online version of this article includes the following source data for figure 4:

**Source data 1.** *Figure 4B*: Proportion of neurons tuned to yaw and pitch.

the feeding task in both subjects (chi-square, p<0.001). In rostral and caudal MIo, 80% of neurons were modulated to 3D direction (bootstrap, p<0.05, *Figure 3B*, left), compared to 52% in areas 1/2 and 3a/3b. Notably, fewer MIo neurons showed directional tuning during swallows compared to chewing, while SIo neurons maintained consistent proportions (*Figure 3—figure supplement 2*; chi-square, MIo: p<0.05, SIo: p>0.1). During drinking, the proportion of directionally modulated neurons was more similar between regions (69% in MIo vs. 60% in SIo: chi-square, p>0.05, *Figure 3B*, right). Mean-matched Fano factor was significantly lower in MIo than SIo in both tasks (Wilcoxon rank-sum test, p<0.001). We considered that the difference in the directional tuning between the two behaviors could be due to the different time intervals used for each task since the period around minimum tongue protrusion in the drinking may contain more of the sensory inputs from the previous lick. However, when sampling spiking activity from an earlier period in feeding, the percentage of directionally tuned SIo neurons was still significantly lower than MIo (chi-square, p<0.001, data not shown). Shifting the period of the drinking trials to be after minimum protrusion also did not lead to a significant difference between MIo and SIo (p>0.1).

Further analysis of the tongue's lateral (yaw) and vertical (pitch) components during feeding revealed additional insights. *Figure 4A* shows peak activity of a neuron in MIo and in SIo at varying degrees of pitch and yaw. Overall, more neurons responded to pitch than yaw (*Figure 4B*), with MIo showing a higher proportion of neurons tuned to both components compared to SIo (chi-square, yaw: p<0.08, pitch: p<0.001), consistent with our 3D direction results. MIo neurons exhibited sharper and narrower tuning curves than the broader tuning curves observed in SIo (*Figure 4A*, see Appendix 2).

## The peak of the PD distribution coincides with leftward tongue movements

The distribution of preferred directions (PDs) provides insight into how tongue muscles are coordinated during movement. Intrinsic and extrinsic tongue muscles are involved in shaping the tongue (e.g. elongation, broadening) and positioning the tongue (e.g. protrusion/retraction, elevation/depression), respectively. These muscles receive bilateral motor innervation except for genioglossus. Straight tongue protrusion requires the balanced action of the right and left genioglossi, while the lateral protrusion involves primarily the contralateral genioglossus. Given this unilateral innervation

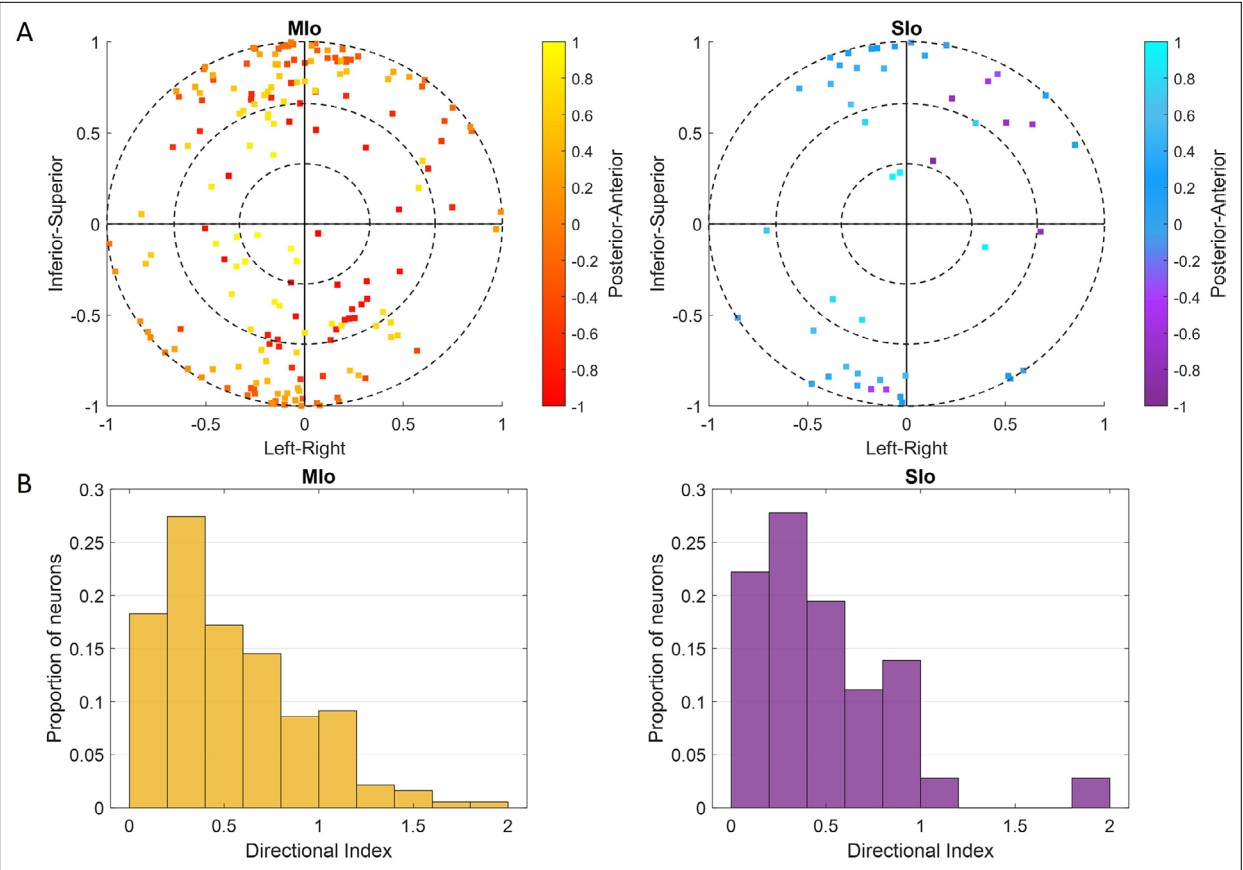

**Figure 5.** Cosine tuning of MIo and SIo neurons. (**A**) Distribution of 3D preferred directions in unit sphere for neurons that fit the tuning function during feeding, combined for both subjects. The origin represents the start of a movement. Color bar represents posterior-anterior axis. (**B**) Distribution of the index for the depth of directional tuning, combined for both subjects (n = 170, 34).

The online version of this article includes the following source data for figure 5:

**Source data 1.** *Figure 5A*: Distribution of 3D preferred directions.

**Source data 2.** *Figure 5B*: Distribution of directional index.

pattern, we hypothesized that left MIo/SIo neurons would preferentially respond to leftward tongue movements, corresponding to right genioglossus activation.

During feeding, MIo and SIo showed nonuniform distribution of PDs across a unit sphere (*Figure 5*; Rayleigh test, p<0.001) and similar directional indices (t-test, p>0.1, mean ± 1 SD: MIo: 0.533±0.3, SIo: 0.604±0.5). As hypothesized, most neuronal populations showed peaks in PD distributions toward leftward tongue movements, except in Monkey R's SIo (*Figure 6A*). Similar results were found with the distributions of preferred yaw during feeding (Appendix 2). While feeding showed comparable PD distributions between MIo and SIo in both subjects (circular k-test, p>0.1), drinking revealed significant differences between regions in Monkey R (chi-square, p<0.001) but not Monkey Y (p>0.09). Monkey Y maintained predominantly left-directed PDs across both tasks, while Monkey R showed more balanced left-right PDs during drinking, suggesting potential involvement of additional muscles beyond the right genioglossus.

## Neural population trajectories differed based on task and cortical regions

To understand how neural circuits, as a dynamical system, perform computations and transform directional information in a high-dimensional state space, we analyzed directional tuning at the population level using factor analysis (FA) to extract neural trajectories and identify patterns in their shared activity. Using the full sample of neurons, the first three factors explained 81% of the variance in the

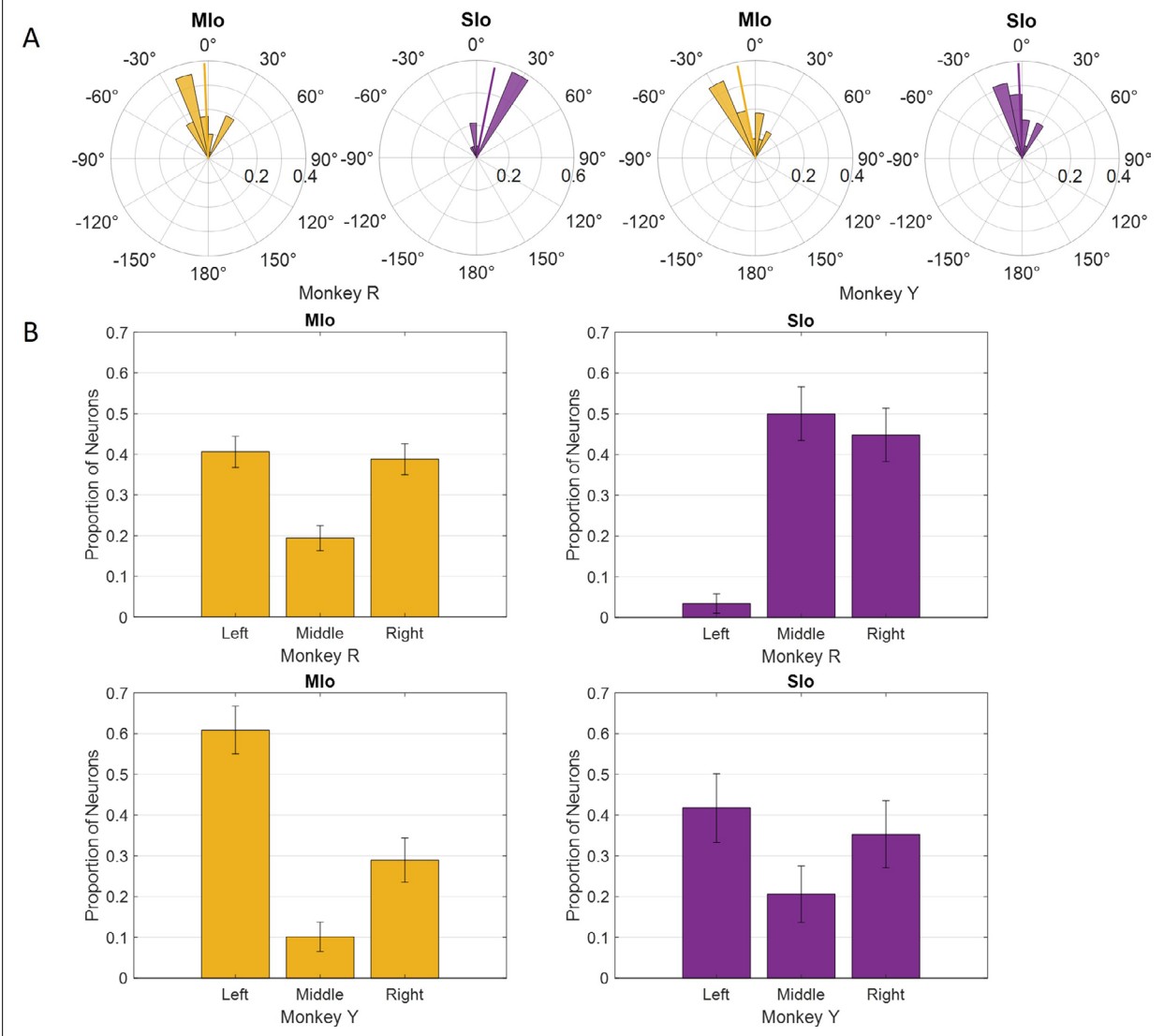

**Figure 6.** Distribution of preferred directions (PDs) in MIo (yellow) and SIo (purple) neurons during control feeding (**A**) and drinking (**B**). For the feeding task, polar plots are split into 10° bins with thick colored lines representing the mean PD. For the drinking task, error bars represent ±1 SE. Neuron counts are included in *Appendix 1—table 1*.

The online version of this article includes the following source data for figure 6:

**Source data 1.** *Figure 6A*: Distribution of preferred directions (PDs) during control feeding.

**Source data 2.** *Figure 6B*: Distribution of preferred directions (PDs) during control drinking.

feeding data (MIo: 82%, SIo: 81%) compared to 71% in the drinking data (MIo: 74%, SIo: 63%). When extended to five factors, feeding sustained higher values with 91% variance explained vs. 82% for drinking. Feeding consistently showed a higher variance explained than drinking across three or five factors, but only three factors were plotted for ease in visualization (*Figure 7*). Similar results were obtained when an equal number of neurons were used (*Figure 7—figure supplement 1*).

To compare population trajectories of feeding and drinking in the same state space, we performed FA on stable neurons across feeding and drinking sessions using all trials (*Figure 7—figure supplement 2A*) and using trials with similar kinematics (*Figure 7—figure supplement 2B*). While the general shape of the population trajectory was preserved across tasks, the inter-trajectory distance between them was significant (t-test, $p<0.001$, mean ± 1 SD: 0.6786±0.0424). Moreover, the MIo trajectory length was longer in feeding (feeding: 0.8954, drinking: 0.6038) despite similar firing rates across tasks (t-test, $p>0.05$) and longer tongue displacement in drinking ($p<0.001$, mean ± 1 SD: feeding:

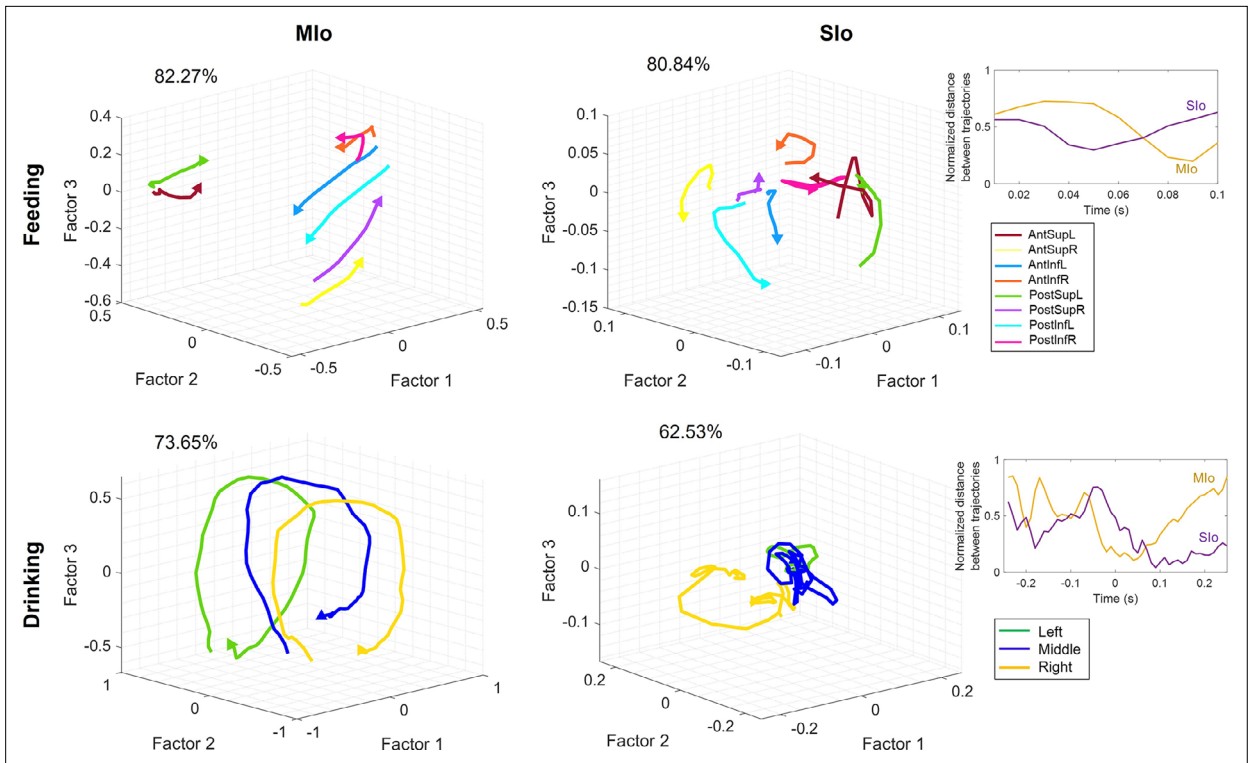

**Figure 7.** Neural population trajectories vary across directions. Trial-averaged trajectories of MIo and SIo population activity along the first three latent factors for Monkey R, grouped by direction. Axes for SIo are 1/4 scale of MIo. Arrows indicate the end of the trajectory. Percentages denote the sum of the variance explained by the first three factors. Inset plots show the difference between the normalized inter-trajectory distances of MIo and SIo over time for both feeding and drinking.

The online version of this article includes the following video, source data, and figure supplement(s) for figure 7:

**Source data 1.** Neural population trajectories.

**Figure supplement 1—source data 1.** Cumulative explained variance for feeding and drinking with equal neurons.

**Figure supplement 2—source data 1.** Comparison between stable MIo neural population trajectories.

**Figure supplement 3—source data 1.** Effect of subsampling on cumulative explained variance.

**Figure 7—video 1.** First three latent variables for trial-averaged neural population trajectories of MIo neurons of Monkey R during control feeding task, grouped by direction.
https://elifesciences.org/articles/101325/figures#fig7video1

**Figure 7—video 2.** First three latent variables for trial-averaged neural population trajectories of MIo neurons of Monkey R during control drinking task, grouped by direction.
https://elifesciences.org/articles/101325/figures#fig7video2

**Figure 7—video 3.** First three latent variables for trial-averaged neural population trajectories of SIo neurons of Monkey R during nerve-blocked feeding task, grouped by direction.
https://elifesciences.org/articles/101325/figures#fig7video3

**Figure 7—video 4.** First three latent variables for trial-averaged neural population trajectories of SIo neurons of Monkey R during nerve-blocked drinking task, grouped by direction.
https://elifesciences.org/articles/101325/figures#fig7video4

**Figure supplement 1.** Comparison of cumulative explained variance between feeding and drinking behaviors with an equal number of neurons (N = 24) for both subjects.

**Figure supplement 2.** Population trajectories of stable neurons.

**Figure supplement 3.** Effect of subsampling (A) equal number of trials per direction (N = 80) from MIo and (B) equivalent neuron counts from MIo and SIo populations (N = 24) on the cumulative variance explained by latent factors.

27.5±9.8, drinking: 47.6±11). When trials were limited to those with similar directional angles (±5°), the difference in trajectory length was no longer observed, but the inter-trajectory distance between tasks remained significantly different (t-test, p<0.001, mean ± 1 SD: 0.6967±0.0793).

In both tasks, neural trajectories of population activity in both MIo and SIo exhibited robust directional information; inter-trajectory distances of all unique direction pairs were significantly higher than zero (*Figure 7*, t-test, p<0.001, for all comparisons in both subjects and region). Notably, in the feeding task, MIo and SIo showed smaller inter-trajectory distances between anterior-posterior paired trajectories during feeding (e.g. AntSupL-brown and PostSupL-green) compared to a greater separation between other directional pairs (*Figure 7*, top, t-test, p<0.05 for both monkeys). Paired superior-inferior directions, e.g., PostSupL-green and PostInfL-cyan, showed the largest separation, though this difference was not significant (p>0.1).

Consistent with previous findings of rotational dynamics in neural population trajectories in nonhuman primates (*Churchland et al., 2012*; *Michaels et al., 2016*; *Russo et al., 2018*), MIo population trajectories in feeding and drinking exhibited rotations that were consistent based on directional components. For example (*Figure 7*, top), trajectories with upward (Sup) components (AntSupL/R, PostSupL/R) rotated opposite from trajectories with downward (Inf) components (AntInfL/R, PostInfL/R). In feeding, Factors 1 and 2 captured superior-inferior and right-left directions, respectively. In drinking, MIo trajectories exhibited consistent rotational direction regardless of spout location (*Figure 7*, bottom left), while exhibiting distinct separation of trajectories for left, center, and right spout-directed tongue movements clustering at approximately –0.5, 0, and 0.5 positions along the Factor 2 axis, respectively. Indeed, inter-trajectory distances in Factor 1 were significantly higher in feeding (t-test, p<0.001, mean ± 1 SD: 0.4628±0.0246) than in drinking (mean ± 1 SD: 0.1286±0.0610), indicating that Factor 1 resembled directional information in feeding but a condition-independent feature of population activity in drinking. The latent factors revealed a clear organizational principle: Factor 1 predominantly captured superior-inferior directional components in feeding, while Factor 2 primarily represented left-right directional components of tongue movement in both tasks.

Similar to previous findings (*Russo et al., 2018*), SIo trajectories in both feeding and drinking showed stark differences from MIo as they were more tangled and exhibited less direct (i.e. sharp turns) paths (*Figure 7*, right). Unlike MIo trajectories, SIo trajectories spanned a smaller neural space, had variable distances between trajectories, and showed inconsistent patterns based on directional components. Quantitative analysis revealed greater separation between normalized trial-averaged population trajectories in MIo compared to SIo (*Figure 7*, inset, t-test, p<0.01, mean ± 1 SD: MIo: 0.44±0.21; SIo: 0.37±0.19). These results were consistent with significantly longer normalized distance traveled by MIo population trajectories compared to SIo in both tasks (Wilcoxon signed-rank test, p<0.05, mean normalized difference in distance traveled ± 1 SD: feeding: 0.18±0.13; drinking: 0.28±0.13). The regional differences cannot be attributed to fewer SIo neurons used (*Figure 7— figure supplement 3*).

## Effects of nerve block

Sensation plays a key role in tongue positioning and movements for natural behaviors. During ingestion, tactile feedback is necessary for locating the bolus, preventing tongue bites, feeling where the drinking spout is, and identifying when it is safe to swallow. To evaluate the role of oral sensation, we used a bilateral oral nerve block to temporarily eliminate tactile sensation in the oral cavity and observe how the control of tongue movement was impacted. Below, we show how the loss of sensation affected both tongue kinematics and directional tuning of neurons during feeding and drinking. To verify that differences between the control and nerve block conditions were due to the loss of sensory feedback and not because of other factors such as sedation and injection, a sham experiment was conducted where saline was administered to the injection sites instead of nerve block. No significant changes to tongue kinematics were observed following the sham experiments (*Figure 8*).

## Changes to tongue kinematics

In feeding, the mean and overall spread of directions were significantly different between the control and nerve block conditions (t-test, p<0.01 and F-test, p<0.001). There was a shift toward a smaller range of 3D directions in Monkey R, whereas there was a shift toward a broader distribution in Monkey Y under the nerve block condition (*Figure 8A*). The positions of maximum protrusion of

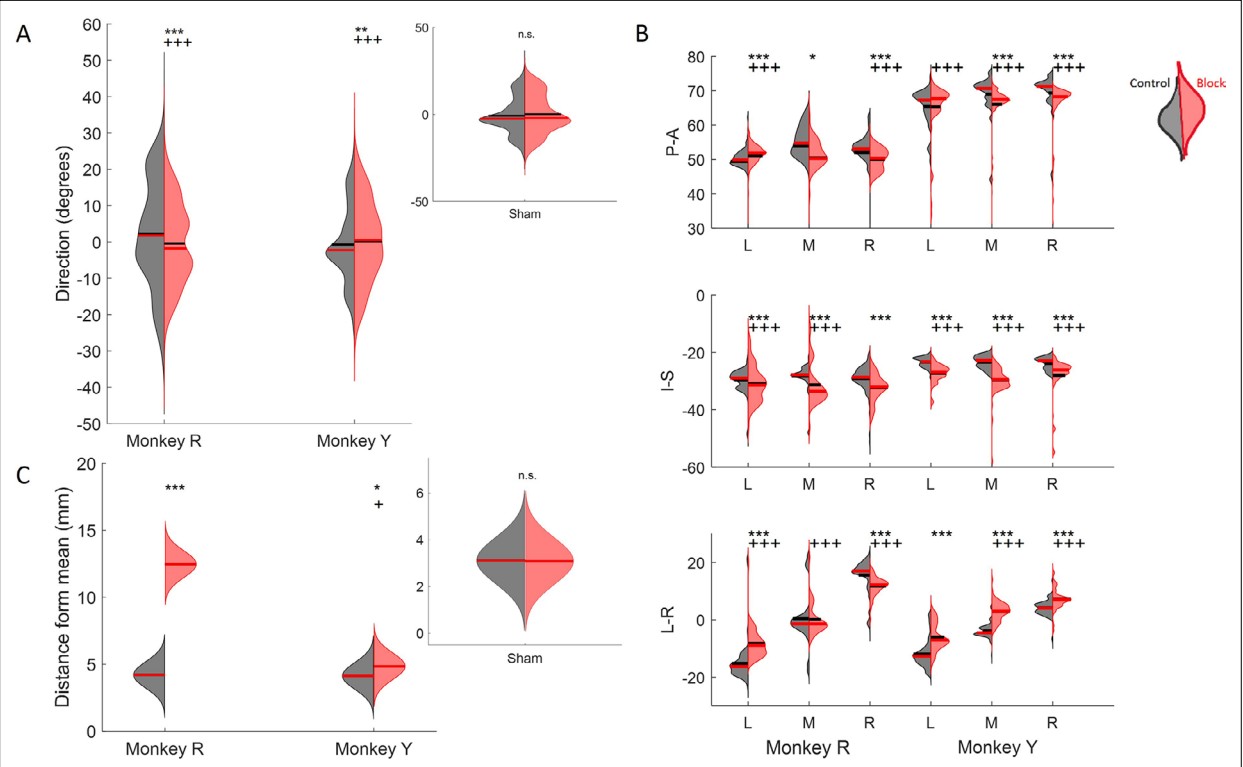

**Figure 8.** Effect of nerve block on direction of tongue movement. (**A**) Distribution of tongue directions during feeding. (**B**) Variance in 3D trajectory endpoints during drinking (posterior-anterior, inferior-superior, left-right) for each direction: left (**L**), middle (**M**), and right (**R**). (**C**) Variation in the distance of drinking endpoint positions from the mean endpoint. Left halves of hemi-violins (black) are control and right halves (red) are nerve block for an individual. Horizontal black lines represent the mean and horizontal red lines the median. Results of two-tailed t-test and F-test are indicated by asterisks and crosses, respectively: *,†p<0.05; **,††p<0.01; ***,†††p<0.001. Smaller inset plots show that there was no effect in the sham nerve block condition, for reference. The sham procedure was identical to the nerve block, except the anesthetic was substituted with saline solution.

The online version of this article includes the following source data for figure 8:

**Source data 1.** *Figure 8A*: Distribution of directions during feeding.

**Source data 2.** *Figure 8B*: Variance in 3D position of drinking trajectory endpoints.

**Source data 3.** *Figure 8C*: Variation in the distance of drinking endpoints from mean.

the tongue during drinking, i.e., the endpoints, were also affected by the loss of sensation. These endpoints represent the planned target position of the tongue to receive the juice reward from a specific spout. In the control drinking task, the endpoints for each spout location were very distinct. In contrast, the endpoints of tongue movements in nerve block exhibited a greater overlap across locations and more variance in all three axes of motion, i.e., posterior-anterior, inferior-superior, and left-right (*Figure 8B*).

Compared to the control, the trajectories of the tongue tip in the nerve block condition during drinking had a smaller range of left-right values. Visually, the tongue trajectories toward the different spout locations were messier and less distinct in failed cycles where the tongue tip missed the location of the correct spout by more than ± 2 SD from the mean (*Figure 9*). In both monkeys, there was a significant increase in the average distance from the mean endpoint position, though this difference was much greater in Monkey R (*Figure 8C*). We noted a difference between subjects in the frequency of failed cycles and the range of left-right tongue movements under nerve block. This may reflect a possible compensatory strategy of reaching the drinking spouts with an adjacent region of the tongue, instead of contacting the right or left spout with the ipsilateral tongue in Monkey R. We observed a decrease in the average speed (t-test, Monkey R: p<0.001, Monkey Y: p>0.1) and an increase in the variance of the speed (F-test, p<0.001) of tongue movement during drinking with nerve block.

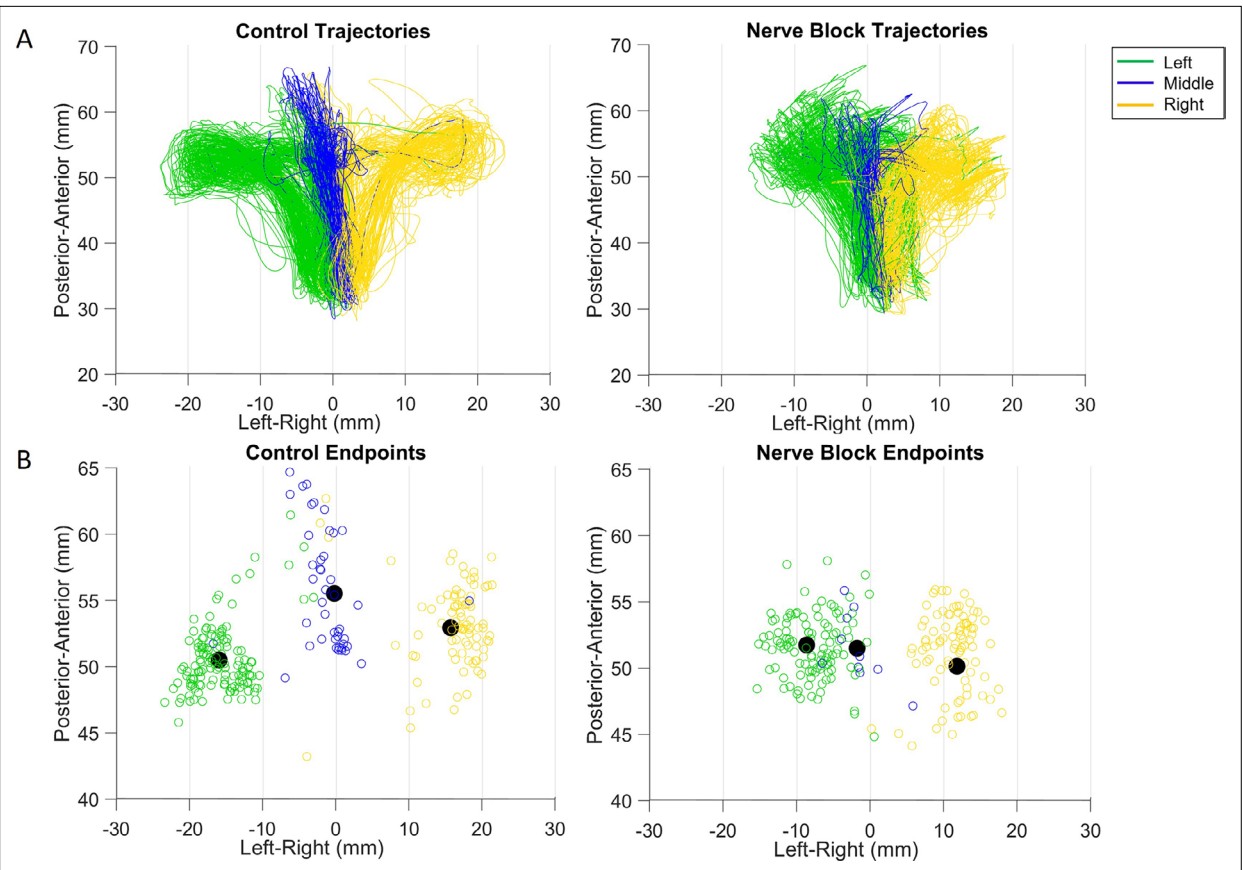

**Figure 9.** Effect of nerve block on drinking kinematics in Monkey R. (**A**) Tongue tip trajectories from starting position to one of three drinking spouts in the control and nerve block conditions. (**B**) Drinking trajectory endpoints, where the black dot represents the mean endpoint position.

## Decreased directional tuning of MIo and SIo neurons

Loss of oral sensation also affected the proportion of directionally tuned neurons and the overall distribution of PDs, though the pattern of changes differed between subjects. Following nerve block, MIo and SIo showed significant decreases in the proportion of directionally modulated neurons across both tasks (*Figure 10A*; chi-square, MIo: $p<0.001$, SIo: $p<0.05$). To confirm this effect was not merely due to altered kinematics, we conducted parallel analyses using carefully subsampled trials with matched kinematic profiles from both control and nerve-blocked conditions. This controlled analysis confirmed the persistent decrease in directional tuning during nerve block (*Figure 10—figure supplement 2*).

We further investigated whether neurons gaining or losing directional selectivity differed across regions. During feeding, MIo and SIo exhibited similar proportions of neurons gaining or losing directional tuning (*Figure 10B*, top row, chi-square, $p>0.1$). The drinking task revealed subject-specific differences (*Figure 10B*, bottom row): in Monkey R, significantly more neurons lost directional tuning in SIo compared to MIo ($p<0.01$), while in Monkey Y, SIo showed a higher proportion of neurons gaining directional tuning than MIo ($p<0.05$). Comparing across behaviors, Monkey R demonstrated consistent patterns of directional tuning changes (chi-square, $p>0.1$), whereas Monkey Y showed significantly higher percentages of neurons gaining directionality during drinking than feeding in both MIo and SIo ($p<0.05$). Interestingly, there was a substantial proportion (40%) of SIo neurons in Monkey Y that gained directional tuning following sensory loss compared to Monkey R (8%) during drinking.

Nerve block significantly altered PD distributions during both tasks. During feeding, MIo neurons in both subjects exhibited a significant clockwise shift in mean PD toward the center (0°), resulting in more uniform distributions (*Figure 11A*; circular k-test, $p<0.01$). In contrast, SIo neurons showed subject-specific responses, with only Monkey R demonstrating a significant counterclockwise shift ($p<0.05$). During drinking under nerve block, MIo neurons displayed subject-dependent directional shifts. In Monkey R, the proportion of neurons with rightward PDs decreased and increased in all

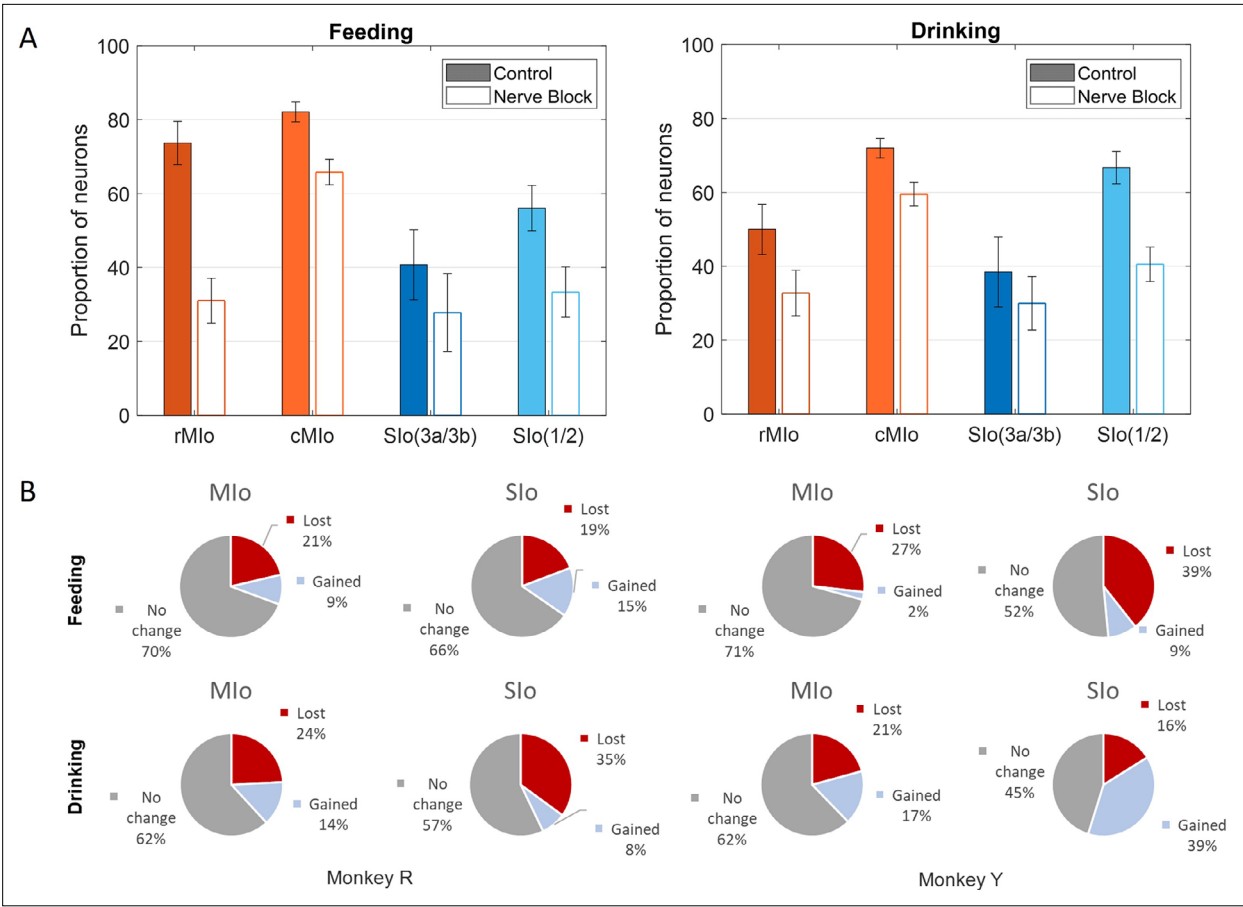

**Figure 10.** Effects of nerve block on directional tuning of orofacial sensorimotor cortex (OSMCx) neurons during feeding and drinking tasks.
(**A**) Percentage of directionally tuned neurons in four areas: rMIo - rostral M1, cMIo - caudal M1, SIo(3a/3b) - area 3a/3b, and SIo(1/2) - area 1/2. Filled-in bars represent control while empty bars represent nerve block. Error bars represent ±1 SE. Neuron counts are included in *Appendix 1—table 1*.
(**B**) Percentage of MIo and SIo neurons which gained or lost directionality with the addition of nerve block.

The online version of this article includes the following source data and figure supplement(s) for figure 10:

**Source data 1.** *Figure 10A*: Percentage of directionally tuned neurons.

**Source data 2.** *Figure 10B*: Percentage of neurons that gained or lost directionality with the addition of nerve block.

**Figure supplement 1.** Effect of subsampling drinking trials with similar kinematic profiles on the change in proportion of directionally tuned neurons in control vs. nerve block conditions for both subjects.

**Figure supplement 1—source data 1.** Effect of subsampling drinking trials with similar kinematics.

**Figure supplement 2.** Change in firing rates of cortical somatosensory neurons.

other directions, whereas Monkey Y showed the opposite with increased neurons with rightward PDs (*Figure 11B*; chi-square, Y: p=0.04). Meanwhile, SIo neurons consistently shifted rightward in both animals (chi-square, R: p=0.02), suggesting differential regional responses to peripheral deafferentation.

## Separation of neural population trajectories was reduced in MIo

Disruption of tactile inputs during feeding had opposite effects on MIo and SIo. Neural population trajectories in MIo showed reduced inter-trajectory distances during nerve block compared to control conditions (*Figure 12*, left, t-test, p<0.05 in >89% of pairs), whereas SIo exhibited increased inter-trajectory distances (*Figure 12*, top right, t-test, p<0.05 in >75% of pairs). In drinking, inter-trajectory distances in both MIo and SIo were significantly reduced across all pairs (two-tailed t-test, p<0.01) except middle-right in Monkey R's SIo neurons (p>0.1).

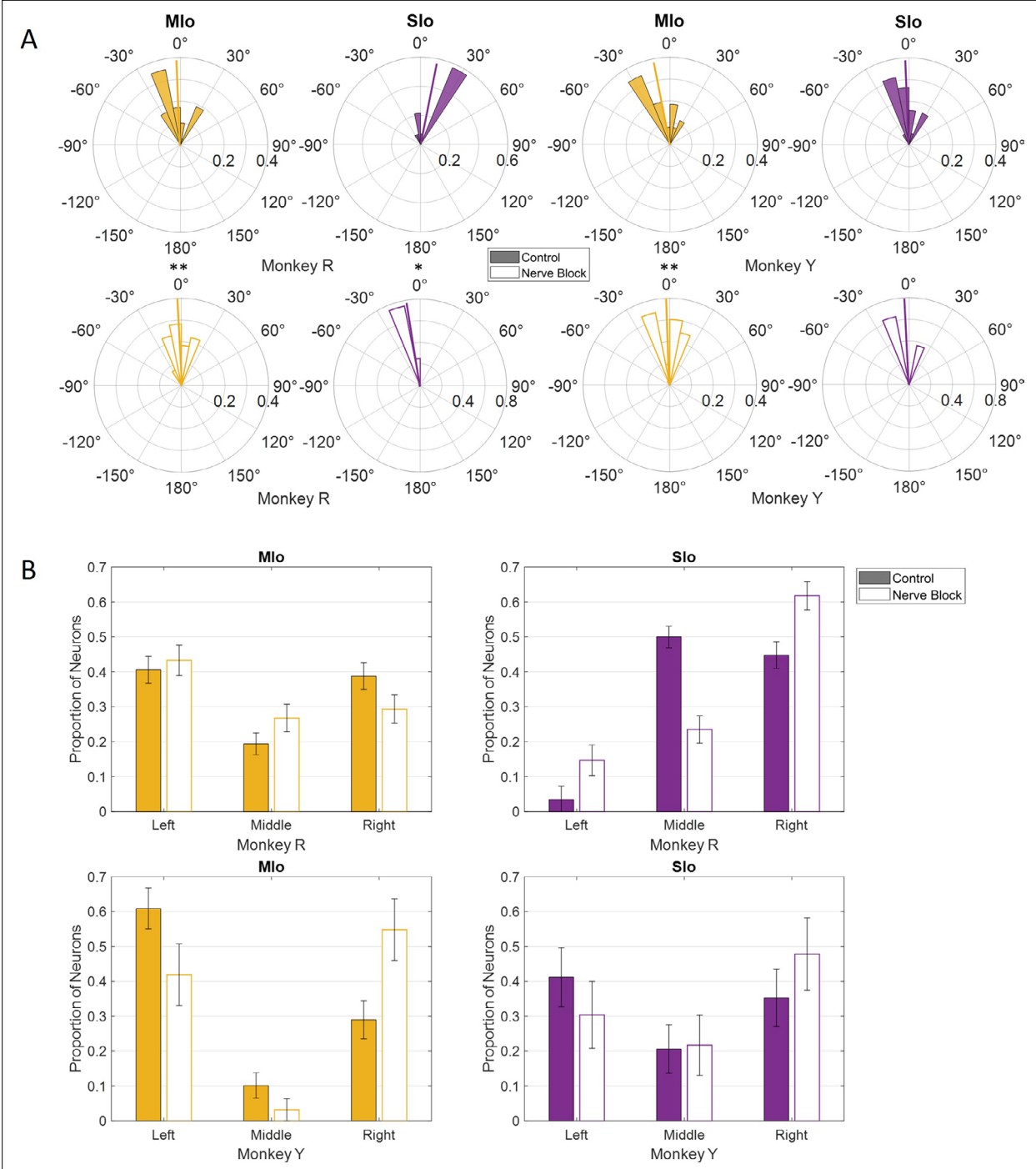

**Figure 11.** Effects of nerve block on the distribution of preferred directions (PDs) of MIo (yellow) and SIo (purple) neurons. (**A**) For the feeding task, polar plots are split into 10° bins with thick colored lines representing the mean PD. Significant circular concentration test (k-test) comparing control and nerve block are indicated by asterisks: *p<0.05; **p<0.01; ***p<0.001. (**B**) For the drinking task, error bars represent ±1 SE. Filled-in bars represent control while empty bars represent nerve block. Neuron counts are included in *Appendix 1—table 1*.

The online version of this article includes the following source data for figure 11:

**Source data 1.** *Figure 11A*: Comparison of preferred left-right directions in control and nerve block during feeding.

**Source data 2.** *Figure 11B*: Comparison of preferred directions in control and nerve block during drinking.

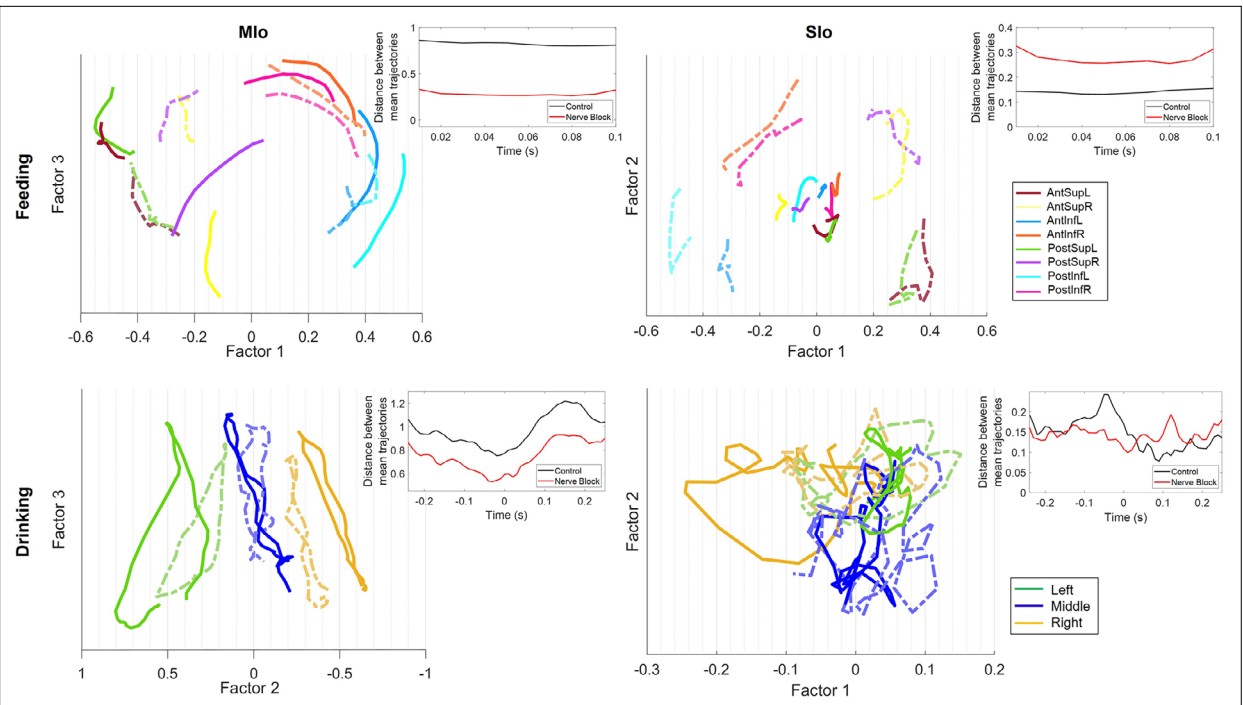

**Figure 12.** Effect of nerve block on population trajectories. Trial-averaged trajectories of MIo and SIo population activity for Monkey R's feeding and drinking sessions, grouped by direction. Two factors with the highest explained variance are plotted for visualization only. Axes represent the latent factors from control data, with the x-axis chosen as the factor with the highest degree of separation between directions. Lighter, dotted lines represent superimposed population trajectories in the nerve block condition. Insets show the difference between the average inter-trajectory distances for control and nerve block conditions.

The online version of this article includes the following source data for figure 12:

**Source data 1.** Effect of nerve block on population trajectories.

The effects of nerve block on the distance traveled by population trajectories were inconsistent across subjects and behavior. Following nerve block, the total distance traveled by SIo trajectories became longer in different behaviors for a specific subject (t-test, Monkey R feeding: p<0.001; Monkey Y drinking: p<0.05). In contrast, MIo trajectories became shorter in drinking (Monkey Y, p<0.01).

## Population decoding of tongue direction

To assess directional information within population activity further, we implemented two decoding approaches to predict tongue movement direction from neuronal spiking patterns: k-nearest neighbor (KNN) classifier and long short-term memory (LSTM) neural network. Consistent with previous study on a cued tongue protrusion task (*Arce et al., 2013*), we found that the 3D direction of tongue movements in naturalistic behaviors could be decoded from simultaneously recorded MIo and SIo populations. The KNN classifier successfully decoded 3D tongue movement direction above chance level across behaviors and experimental conditions (*Figure 13A*). Results of analyses using multiple linear regression model with interactions revealed several key factors affecting decoder performance: behavior type (p<0.001, drinking outperformed feeding by 11%), cortical region (p<0.001, MIo exceeded SIo by 13%), and inter-subject variability with behavior (p<0.001, 12% higher for Monkey R during drinking, comparable accuracy during feeding). Notably, disrupting tactile sensation through nerve block did not significantly impair KNN classifier performance (p>0.1). The results using LSTM were different from that of KNN; decoding tongue direction using LSTM showed substantially higher performance when using MIo neural activity (mean $R^2$ ±1 SD: control: 0.46–0.81±0.1, nerve block: 0.26–0.7±0.1) compared to SIo (mean $R^2$ ±1 SD: control: 0.12–0.43±0.1, nerve block: 0.05–0.17±0.07) across all experimental conditions (*Figure 13B*, t-test, p<0.001). This regional difference became particularly pronounced during nerve block, where SIo decoding accuracy decreased substantially more than MIo, suggesting differential reliance on tactile feedback

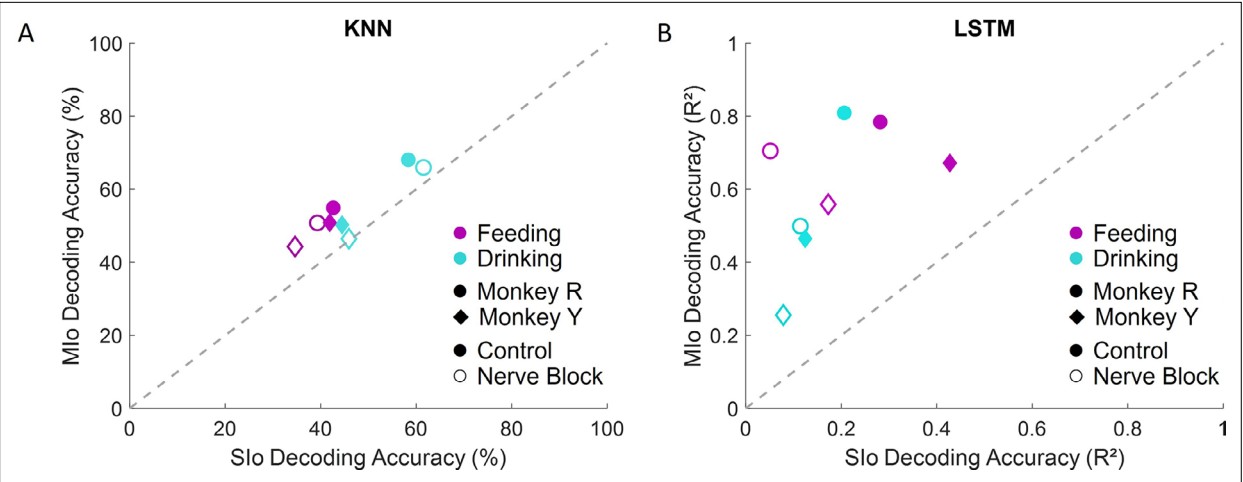

**Figure 13.** Accuracies of two different decoding algorithms from MIo and SIo populations of equal size (N=28). (**A**) Comparison between average decoding accuracy of k-nearest neighbor (KNN) classifier. Chance level is 33.33%. (**B**) Comparison between average decoding accuracy by long short-term memory (LSTM) network. Data are shown separately for each subject, behavioral task, and condition. The dashed line signifies equal decoding performance for MIo and SIo. Decoding accuracies from full populations are included in *Figure 13—figure supplement 2*.

The online version of this article includes the following source data and figure supplement(s) for figure 13:

**Source data 1.** *Figure 13A*: k-Nearest neighbor (KNN) decoding performance.

**Source data 2.** *Figure 13B*: Long short-term memory (LSTM) decoding performance.

**Figure supplement 1.** Correlation between population size and decoding accuracy.

**Figure supplement 2.** Decoding accuracies from neuronal populations of various sizes.

**Figure supplement 2—source data 1.** k-Nearest neighbor (KNN) decoding accuracies from neuronal populations of various sizes.

between these cortical regions. Combining MIo and SIo showed significantly better decoder performance compared to performance using neuronal populations separately (mean $R^2$ ±1 SD: control: 0.78–0.92±0.05, nerve block: 0.58–0.91±0.05, p<0.001) in feeding, but not drinking. To address the potential confound of varying population sizes between MIo and SIo, we standardized comparisons by downsampling all populations to match our smallest recorded group (N=28 neurons). Decoding accuracy improved by up to 10% when using all neurons in MIo or SIo compared to using subsamples of neurons. Decoding using LSTM showed consistently higher accuracies in feeding compared to drinking regardless of the length of intervals used (100 ms, 500 ms), behavioral window (±500 ms relative to minimum protrusion, between maximal protrusions), and directional angles (actual vs. {–45, 0, 45}). These results suggest that continuous and nonlinear decoding is better suited for feeding than drinking behavior.

## Discussion

Our study investigated the representation of 3D directional information in MIo and SIo during natural feeding and drinking behaviors, and how cortical representations change when tactile sensation is disrupted. By simultaneously recording large-scale cortical activity and 3D tongue kinematics, we revealed a nuanced neural encoding of tongue movement direction that varies systematically across cortical regions, behaviors, and sensory feedback conditions. We found that a substantial proportion of neurons exhibit directional tuning characterized by diverse tuning curve properties (PD, tuning shape, modulation depth). Neural population trajectories demonstrated distinct patterns across different movement directions. Directional tuning at both individual and population levels is more robust in MIo. Following sensory loss, alterations in tongue kinematics were accompanied by changes in directional information in MIo and SIo, manifesting as modifications in both individual neuron tuning characteristics and the broader dynamics of population-level neural activity.

## Differences across behaviors

In the present study, results from the more natural drinking are consistent with previous findings that MIo and SIo encode tongue direction during a trained protrusion task (*Murray and Sessle, 1992*; *Sessle et al., 2005*; *Arce et al., 2013*). Our study extends this knowledge by investigating the dynamics of directional tuning of individual and population of neurons in OSMCx to 3D tongue direction during naturalistic behaviors. Unlike previous similar studies, the monkeys were not trained to reach specific targets and were instead allowed to eat and drink relatively naturally. By comparing two naturalistic behaviors, we found that the directional information in OSMcx was higher in feeding than in the drinking task, as seen in higher proportion of directionally tuned neurons, cumulative variance explained by latent factors, and decoding accuracies. Strikingly, fewer latent factors explain most of the variance in the feeding data, suggesting that the underlying network activity is relatively simple despite apparent complexity of feeding.

That directional tuning is modifiable is consistent with previous findings in primate motor cortex where directional tuning was modulated by movement parameters such as speed, posture, distance (*Aflalo and Graziano, 2006*), and by varying task contexts such as availability of prior information (*Rickert et al., 2009*), individual vs. segmented arm movements (*Ben-Shaul et al., 2004*), one- vs. two-target reaching (*Ebina et al., 2024*). A high degree of similarity in neural modes has been reported across different wrist tasks in 1D and 2D (*Gallego et al., 2018*). This suggests that in our study, feeding and drinking may reflect more distinct biomechanical constraints and sensorimotor requirements compared to the wrist tasks. In feeding, the tongue moves in varied directions to position the food between left-right tooth rows during chewing, whereas in the drinking task, the tongue moves to discrete and fixed spout locations. Additionally, feeding and drinking engage the jaw differently. During feeding, the jaw moves more extensively and in multiple directions, while jaw movements during drinking are smaller and primarily vertical. Lastly, the tongue-jaw coordination differs between tasks; maximum tongue protrusion coincides with maximum gape in drinking but with minimum gape in the feeding behavior (*Laurence-Chasen et al., 2022*; *Punacha et al., 2024*). Indeed, MIo trajectories' predominant latent factor contained directional information in the feeding task, resembling superior-inferior directional components which rotated in opposite directions. In contrast, MIo trajectories in drinking exhibited a consistent rotational direction regardless of spout location (*Figure 7*). This may reflect a predominant nondirectional information such as condition-independent time-varying spiking activity during drinking (*Kaufman et al., 2016*; *Kobak et al., 2016*; *Arce-McShane et al., 2023*).

## Comparison between MIo and SIo

Directional tuning of neurons during feeding showed a notable disparity between MIo and SIo, suggesting that MIo carries more robust directional information for tongue movements in feeding tasks. The similar proportions of directionally tuned MIo and SIo neurons in the drinking task studied here were consistent with previous findings (*Arce et al., 2013*). At the population level, MIo trajectories showed more consistent rotational patterns and greater inter-trajectory separation than those in SIo. When neural population trajectories are farther away from each other in state space, it means that the patterns of activity across tongue directions are more distinct and separable, thus less likely to be confused with each other. This may signify that neural representations of 3D tongue directions are more robust. When there is better neural discrimination and more reliable information processing, it is easier for downstream brain regions to distinguish between different tongue directions.

Consistent with results from previous studies (*Michaels et al., 2016*; *Seely et al., 2016*; *Russo et al., 2018*), MIo trajectories exhibited low tanging and smoother dynamics than SIo trajectories. These may suggest that the low tangling in MIo confers noise robustness while the higher tangling in SIo reflects variability in the tactile signals received by SIo during feeding. Consistent with our previous study (*Laurence-Chasen et al., 2023*), decoding from MIo yielded higher accuracies than from SIo in both behaviors. These results support the well-established role of MIo in the control of movement parameters, especially direction. Varying tongue shape, tongue's contact with varying bolus properties (size and texture), and other oral structures (palate, teeth) may weaken the directional signal contained in SIo activity. Thus, slight differences in tongue kinematics might create significant differences in sensory signals across trials. When looking at trial-averaged signals, this natural variability could make the neural response patterns appear less precise or specific than they are. These

are consistent with our findings that for both tasks, spiking variability was higher in SIo, and that the variance accounted for was lower in SIo population activity compared to MIo.

## Laterality in OSMCx

Similar to previous results in arm motor cortex (*Lillicrap and Scott, 2013*), we observed nonuniform PD distributions consistent with the frequency distribution of tongue movements, suggesting that neural populations contain information that reflects the anatomical constraints of the tongue. The highest frequency of both observed directions and directional tuning peaks was in the anterior and superior directions. We additionally found that the peak of the PD distribution, especially in feeding, coincides with leftward tongue movements, suggesting the presence of laterality in the PDs of OSMCx neurons. Previous results in humans examined using fMRI reported that hemispheric differences in sensorimotor activity during voluntary tongue movements are related to the preferred chew side (*Shinagawa et al., 2003*). This was not the case in our study as the preferred chew side was the same for both monkeys (i.e. right side), despite differences in the predominant PD. It is possible that the difference between the two subjects is related to the difference in recording locations, with Monkey Y's being more lateral and therefore closer to the swallow area of the cortex than Monkey R's (*Figure 2—figure supplement 2*). Monkey Y had a higher proportion of neurons that were tuned to tongue direction during feeding compared to Monkey R (*Figure 3—figure supplement 2*), but fewer during drinking. An avenue for further study could be a unilateral nerve block on the preferred side to measure how the unaffected side of the tongue compensates for the lack of sensation in the affected side. A previous study found that unilateral lingual nerve transection in pigs alters the coordination of the ipsilateral tongue side during chewing (*Montuelle et al., 2020*). The tongue is a complex group of muscles, with intrinsic muscles primarily contributing to the shape of the tongue and extrinsic muscles contributing more to the positioning of the tongue. Therefore, it is possible that the neurons which are strongly tuned to tongue direction have direct connections to the extrinsic muscles on the ipsilateral side. Looking at how each side of the tongue responds independently to unilateral nerve block, and how this interacts with directional preference may give us more information about how the unique structure of the tongue is coordinated.

## Role of tactile feedback

Previously, we reported that the administration of bilateral nerve block to the sensory branches of the trigeminal nerve impaired feeding performance and tongue-jaw coordination (*Laurence-Chasen et al., 2022*). The present study extends these findings by showing that directional movement of the tongue (*kinematics*) and the directional information in both MIo and SIo are also affected by the loss of tactile inputs from the tongue and other structures of the oral cavity (e.g. palate, teeth, gingiva). These findings highlight the critical role of sensory information in sensorimotor control in general (*Dadarlat et al., 2015*; *Delhaye et al., 2018*) and in the representation and computation of directional signals for controlling tongue movements. MIo and SIo neurons, which respond to tactile and proprioceptive inputs from the tongue (*Huang et al., 1989*; *Lin et al., 1994*; *Toda and Taoka, 2002*; *Toda and Taoka, 2004*; *Toda and Taoka, 2006*; *Arce et al., 2013*; *Cerkevich et al., 2014*), use sensory information to plan and adjust tongue movements to achieve contact with the spout or position the bolus appropriately at different stages of the feeding sequence. Without tactile feedback, subjects may rely on alternative sensory cues like taste for locating the spout or bolus (*Todrank and Bartoshuk, 1991*). In a recent optogenetic inhibition study on licking in mice, it was found that the tongue/jaw regions of the somatosensory cortex were necessary for proper tongue targeting but not for the core motor capabilities of the tongue (*Xu et al., 2022*). The reduced range of tongue motion we observed likely stems from sensory loss rather than motor impairment. While our experimental setup did not eliminate visual feedback that monkeys might use to readjust tongue position in the drinking task, oral sensory loss alone had a significant effect on the monkeys' performance in feeding, as the tongue was out of sight within the oral cavity.

Individual differences were notable in our study, possibly due to differences in the electrode array placement or compensatory strategies. This individual variability suggests that studying additional subjects would provide valuable insights into how OSMCx adapts following sensory disruption. In our study, head position and hand movement were restrained to eliminate contributions of hand-to-mouth movements when handling food or drink. The hand and orofacial cortical areas are

anatomically adjacent and highly interconnected (*Forrester and Rodriguez, 2015*), and researchers have found a neural region in mice that coordinates hand-to-mouth movements during natural feeding (*An et al., 2022*). Truly natural feeding would involve holding food up to the mouth, as well as free head movement, which would make tracking of the marker positions difficult under this experimental setup. Advances in tracking tongue movements would be necessary to study more complex feeding sequences.

## Clinical implications

This study offers new information about the important role of sensorimotor integration in controlling tongue direction during natural behaviors. There is a high degree of directional information contained in the spiking activity of the orofacial cortex, especially in the motor areas. The bilateral nerve block's effect enhances our understanding of the processes affected by oral sensorimotor dysfunctions such as trigeminal neuropathies. It demonstrates the importance of oral sensation for supporting the full range of directional motion but also shows that significant directional information can be extracted even in the absence of tactile feedback. This type of knowledge can inform the diagnosis and rehabilitation of orolingual dysfunctions, following stroke or glossectomy. There have been advancements in brain-computer interface (BCI) by decoding the real-time signals of arm region of the motor cortex to control prosthetic arm movement (*Collinger et al., 2013*; *Vilela and Hochberg, 2020*) or muscle stimulation (*Ethier and Miller, 2015*; *Canny et al., 2023*), as well as efforts to restore sensory feedback by stimulating correct areas of somatosensory cortex in response to sensors on a prosthetic (*Tabot et al., 2013*; *Flesher et al., 2021*). Recently, there has also been progress in speech BCIs, which utilize recordings from sensorimotor areas to restore communication abilities (*Silva et al., 2024*; *Stavisky, 2025*). That the OSMCx, particularly MIo, can rapidly decode tongue direction during natural behaviors is significant for developing neuroprosthetic control or soft prosthetics.

## Methods

### Subjects

Experiments were performed on two adult male rhesus macaques (*Macaca mulatta*, 9–10 kg, ages 8 and 9 years) in the University of Chicago XROMM Facility. This sample size was chosen based on precedent in the field of nonhuman primate motor neuroscience. Experiments were performed in the University of Chicago XROMM Facility. The protocol (#72556) was approved by the University of Chicago Animal Care and Use Committee (IACUC) and complied with the National Institutes of Health Guide for the Care and Use of Laboratory Animals. The subjects were seated in a standard primate chair and head-fixed to keep their head position constant during feeding and drinking trials. Each trial lasted 10 s. In a feeding trial, a piece of food (grape, gummy bear, pasta) of roughly the same size was presented directly to the animals' mouth using a stylus. In a drinking trial, juice was delivered through one of three spouts positioned in front of the subject (*Figure 1A*).

### Cranial nerve V anesthesia

For some sessions, these behavioral tasks were preceded by nerve block injections (0.25% Bupivacaine HCL and Epinephrine 1:200,000, 0.25 mL/injection site) to the sensory branches of bilateral trigeminal nerves (lingual, inferior alveolar, buccal, palatine) to eliminate oral tactile sensation locally and temporarily. The nerve block was administered while the subjects were under sedation, and all data were collected within 90 min of the nerve block. Each monkey served as its own control, with nerve block feeding data collection sessions taking place either a day before or a day after the associated control session. Nerve block drinking data collection was performed immediately following the control drinking session. Multiple datasets (40–60 trials) were collected for both subjects across multiple days. However, due to the complex and time-consuming nature of processing integrated X-ray Reconstruction of Moving Morphology (XROMM) and neural data, one session per subject, behavior, and condition was used for this study. Thus, we analyzed a total of eight datasets.

### Video-radiography

Prior to data collection, the animals were implanted with spherical tantalum beads (1 mm diameter) in the cranium, mandible, and the tongue, from the tip to the region of the circumvallate papillae.

During feeding or drinking, the movement of these markers was recorded using high-resolution (200 Hz, <0.1 mm) biplanar video-radiography collected with Xcitex ProCapture version 1.0.3.6. The 3D positional data was obtained following the previously described XROMM workflow (*Laurence-Chasen et al., 2020*) incorporating the use of XMALab (*Knörlein et al., 2016*) and machine learning using DeepLabCut (*Mathis et al., 2018*) to reconstruct the kinematic data. The x, y, z values of the markers were then smoothed with a 30 Hz low-pass Butterworth filter and transformed into a cranial coordinate space with the origin fixed at the posterior nasal spine. Gape cycles within each feeding sequence were manually identified and categorized by cycle type (manipulation, stage 1 transport, chew, stage 2 transport, or swallow).

## Electrophysiology

Under general anesthesia, a microelectrode array was chronically implanted in four areas of the left hemisphere (*Figure 2—figure supplement 2*): rostral MIo (96-electrode Utah array; electrode length: 1.5 mm), caudal MIo (32-electrode floating microelectrode array [FMA], electrode length: 3.0–4.5 mm), area 1/2 (96-electrode Utah array, electrode length: 1.0 mm), and area 3a/3b (32-electrode FMA, electrode length: 4.0–8.7 mm). The neural data was recorded using Grapevine Neural Interface Processor (Ripple Neuro, Salt Lake City, UT, USA). Signals were amplified and bandpass filtered between 0.1 Hz and 7.5 kHz and recorded digitally (16-bit) at 30 kHz per channel. Only waveforms (1.7 ms in duration; 48 sample time points per waveform) that crossed a threshold were stored and offline spike sorted (Offline Sorter, RRID:SCR_000012) to remove noise and to isolate individual neurons. Neurons recorded during control feeding sessions are the same as those previously reported in *Laurence-Chasen et al., 2023*. The channel name assigned to each recorded neuron was kept consistent between control and nerve block data for comparison.

## Data analysis

### 3D kinematics

3D tongue kinematics were recorded simultaneously with the neural data in all behavioral sessions. All data analyses were performed in MATLAB 2022b (RRID:SCR_001622). For feeding, the instantaneous 3D direction of the tongue tip marker for every 100 ms throughout each gape cycle was calculated as:

$$3D\,angle,\ \theta = tan^{-1}(\| v_1 \times v_2 \| / v_1 \cdot v_2) \tag{1}$$

where $v_1$ is the x, y, z position at the start of each 100 ms interval and $v_2$ is the position at the end (*Figure 1B*). These directions were then categorized based on whether the movement was negative or positive relative to the horizontal plane (left/right), the sagittal plane (inferior/superior), and the x axis (posterior/anterior). This resulted in eight directions: AntSupL, AntSupR, AntInfL, AntInfR, PostSupL, PostSupR, PostInfL, and PostInfR. An equal number of 100 ms intervals from each of these directions was sampled to eliminate the possible effect of different distributions of kinematics across datasets, and spike data during each interval was used for neural analysis. For comparison with the drinking task, the sign was determined relative to the horizontal plane, with rightward tongue movement being positive. This is also the plane of motion which has been the least studied. These left-right directions were categorized into six 10°-bins with a total range of –30° to 30°, which encapsulated most of the observed distribution of directions in each subject. Lingual yaw (transverse rotation) and pitch (elevation/depression) were also calculated to compare tuning across the lateral and vertical components of tongue direction (Appendix 2, *Equation 2*). For drinking, the direction was determined by which of the three spouts juice was dispensed from during each lick. The spiking activity used for neural analysis of the drinking task was from intervals of ±250 ms around each minimum protrusion of the tongue. As 100 ms was not sufficient to capture the full range of tongue motion during each drinking cycle, the length of time used was increased to allow a clear distinction between the three directions. The period of 250 ms spans about 75% of the average lick length from minimum to maximum protrusion of the tongue.

Kinematic performance for feeding was determined by the spread of tongue directions observed across trials. For drinking trials, performance was determined by the variance of endpoint positions, as well as by the proportion of 'failed' cycles, where the monkey missed the correct spout location with their tongue tip. The difference between control and nerve block performance was evaluated using a two-tailed t-test and F-test.

## Directional tuning of single neurons

Tongue directions were subsequently compared with the firing rates of individual neurons across cortical areas. To determine if neurons were directionally modulated, we used a bootstrap procedure (*Arce et al., 2010*): we resampled the firing rates from an equal number of trials in each direction with replacement 1000 times and computed 95% confidence intervals from the resulting distribution to test whether the mean ranks are the same across directions. The proportions of neurons found to be directionally tuned were compared across groups using a chi-square test. Due to limited neuron counts in some cortical regions, we combined rM1 and cM1 recordings as MIo, and areas 1/2 and 3a/3b as SIo for subsequent analyses (*Appendix 1—table 1*). Then, multiple linear regression was used to determine if the firing of each neuron fits the cosine tuning function that has been previously described for the arm area of the motor cortex (*Schwartz et al., 1988*). To accomplish this, the directional components of a unit vector representing each group of directions were calculated. For neurons that fit the tuning function, a PD in 3D space was estimated. These PDs are distributed around a unit sphere, with the origin representing the start of the movement. The directional index was calculated as a measure of the depth of directional tuning. To determine PDs for the drinking task, we resampled the original distribution of firing rates with replacement for each direction and calculated the direction for which a neuron exhibited its maximal firing rate over 1000 bootstrap samples. Similarly, a PD across the left-right feeding directions was determined for comparison. Circular concentration (k-test) to compare distributions of PDs during feeding and polar plot generation was performed using the CircStat MATLAB toolbox (*Berens, 2009*). For drinking, distributions of PDs were compared using a hi-square test. We analyzed the trial-by-trial variability of neuronal activity using the Fano factor, which was computed as the spike count variance divided by spike count mean within each session. The Fano factor was calculated separately for each subject, task, and cortical region. For analysis across sessions, we used the mean-matched Fano factor (*Churchland et al., 2010*).

## FA of population activity

We used FA, a linear dimensionality reduction method, to obtain latent trajectories of spiking activity and compare population responses to different directions across trials. FA is defined as:

$$y \sim n(\mu, CC' + R) \tag{2}$$

where y is the spike counts from n neurons, μ is the mean spike counts from **n** neurons, C (m × n) is the loading matrix mapping **m** latent factors to the spike counts of **n** neurons, and R (n × n) represents the unexplained variance of independent neurons (*Santhanam et al., 2009*; *Horrocks et al., 2024*). The parameters μ, C, and R were estimated with expectation-maximization using DataHigh MATLAB toolbox (*Cowley et al., 2013*). The latent factor space obtained using FA represents the shared population variance of neurons. To obtain latent factors of shared population activity, we binned the spike times of each neuron into 10 ms bins (10 bins for feeding; 50 bins for drinking). Neurons with a mean firing rate <1.0 spike/s were excluded, and the resultant vectors were smoothed using a Gaussian kernel with a 10 ms standard deviation (SD). To determine the dimensionality of the latent variable, we used threefold cross-validation to find the value of **m** which maximized the likelihood of the data. We then obtained an FA model by fitting an m-dimensional latent factor model. FA was performed using trial-averaged data to examine the direction-relevant latent factor responses (i.e. neural population trajectories).

We quantified directional differences in population activity by calculating the Euclidean distance over all the latent factors (m=20) between trial-averaged neural population trajectories for each unique direction pair (drinking = 3 pairs; feeding = 28 pairs). This analysis was performed for every 10 ms bin throughout each trial. Since the latent spaces were derived separately for MIo and SIo, we compared their normalized mean inter-trajectory distances obtained by first calculating the geometric index (GI) of the mean inter-trajectory distances, d, between each pair of population trajectories per region as: GI = (d$_1$–d$_2$)/(d1+ d2). To assess differences between experimental conditions (control vs. nerve block) or cortical region (MIo vs. SIo), we applied a two-sample t-test to the mean inter-trajectory distances across all direction pairs. We further characterized the neural space spanned by population activity by measuring the cumulative Euclidean distance traveled by trajectories from start to end of a trial.

To control for potential sampling biases, we implemented two critical validation procedures. First, we addressed the varying trial counts across directions in the feeding task by performing FA with

standardized samples (N=80 trials per direction) through random subsampling repeated 10 times. We then compared the cumulative explained variance between the full and subsampled datasets. Second, we controlled for population size differences by subsampling MIo and SIo neurons to equivalent counts, enabling unbiased comparison of FA results between cortical regions.

To compare behaviors, we performed FA on only caudal MIo neurons that remained stable across both feeding and drinking recording sessions (N=20). These neurons were determined through a stability test (*Dickey et al., 2009*), which compared the average waveform and interspike interval for both datasets. We analyzed two subsets of kinematic data: data collected within 200 ms surrounding minimum gape, and data from trials where the 3D angle measured 100 ms after minimum tongue protrusion was between –5° and +5°.

### Decoding tongue direction

The ability to predict tongue direction from spiking activity of MIo and SIo neurons was evaluated using a KNN classifier. The Euclidean distance was used to identify nearest neighbors, and the number of nearest neighbors used was K=7. This K value was determined after testing different Ks which yielded comparable results. The feature was the z-scored firing rate of each neuron over each trial: every 100 ms throughout feeding sequence, or 100 ms centered at minimum tongue protrusion during drinking. As a more direct comparison to the drinking, feeding directions were split into three groups representing left, middle, and right movement directions. The decoder was trained on 80% of trials and tested on the remaining 20%, then decoder performance was determined by the percentage of test trials where the direction of movement was correctly decoded from the neural data. We ran 100 iterations of the classifier using a different set of randomly selected training and test trials, then calculated the average performance. The same sets of training and test trials were used for decoding from simultaneously recorded MIo and SIo data. However, our recorded populations were of variable sizes, and decoding performance was found to be related to the number of neurons in the ensemble (*Figure 13—figure supplement 1*). Because the smallest population of neurons we recorded was 28, we selected 28 random neurons from the larger populations for each iteration. Based on the positive relationship between population size and decoding accuracy, we expect that performance would increase with more neurons. These results will show whether tongue direction can be decoded from a small number of neurons. We fit a linear regression model with interactions to compare decoding performance across the other variables in the experiment. To determine if a mixed population of MIo and SIo neurons performs better than the pure populations, we started with the full MIo population and systematically replaced 25 MIo neurons with an equal number of SIo neurons. We repeated this replacement over 100 iterations, each with a different random selection of neurons, and decoded tongue direction using the KNN classifier. We compared the average decoding performance of these slightly different mixed populations to the baseline of running the decoder with the full MIo population.

We also decoded tongue direction using an LSTM network (*Hochreiter and Schmidhuber, 1997*; *Glaser et al., 2020*; *Laurence-Chasen et al., 2023*; *Hahn and Arce-McShane, 2024*) implemented in MATLAB's Deep Learning Toolbox. For the feeding task, we analyzed 3D tongue movement direction (relative to the sagittal plane) at 100 ms intervals. For the drinking task, we categorized tongue direction every 500 ms as either left (–45°), middle (0°), or right (45°). For each neural population, we created a 2D array containing z-scored spike counts for these time intervals (neurons × intervals). We randomly selected five groups of 28 neurons with replacement from each population. Using five-fold cross-validation, we trained an LSTM network (400 hidden units, 50 training epochs) on 85% of the intervals. The network was then tested on the remaining 15% of neural data in a stepwise manner, producing a sequence of predicted tongue directions. We used mean $R^2$ to measure prediction accuracy.

## Acknowledgements

We thank JD Laurence-Chasen for data collection and software, Christina Hahn for assistance with LSTM analysis, as well as all members of the Arce-McShane Lab past and present, including Rebecca Junod, Hernando Ferreira, Derrick Tang, Emma Lesser, Jared Luckas, Tricia Nicholson, and Eric Hosack, for assistance with data collection and processing. This research was supported by National Institutes of Health grants from the National Institute on Aging, Grant Number: R01AG069227 (to FIA-M, PI)

and the National Institute of Dental and Craniofacial Research, Grant Number: R01DE027236 (to FIA-M, PI). The content is solely the responsibility of the authors and does not necessarily represent the official views of the National Institutes of Health. The funders had no role in study design, data collection, and interpretation, or the decision to submit the work for publication.

## Additional information

### Funding

| Funder | Grant reference number | Author |
|---|---|---|
| National Institute on Aging | R01AG069227 | Fritzie Arce-McShane |
| National Institute of Dental and Craniofacial Research | R01DE027236 | Fritzie Arce-McShane |

The funders had no role in study design, data collection and interpretation, or the decision to submit the work for publication.

### Author contributions

Victoria B Hosack, Formal analysis, Visualization, Writing - original draft, Writing - review and editing; Fritzie Arce-McShane, Conceptualization, Supervision, Funding acquisition, Investigation, Methodology, Project administration, Writing - review and editing

### Author ORCIDs

Victoria B Hosack ⓘ https://orcid.org/0000-0001-5415-584X
Fritzie Arce-McShane ⓘ https://orcid.org/0000-0001-6616-3564

### Ethics

Experiments were performed in the University of Chicago XROMM Facility. The protocol (#72556) was approved by the University of Chicago Animal Care and Use Committee (IACUC) and complied with the National Institutes of Health Guide for the Care and Use of Laboratory Animals.

Reviewer #1 (Public review): https://doi.org/10.7554/eLife.101325.3.sa1
Reviewer #2 (Public review): https://doi.org/10.7554/eLife.101325.3.sa2
Reviewer #3 (Public review): https://doi.org/10.7554/eLife.101325.3.sa3
Author response https://doi.org/10.7554/eLife.101325.3.sa4

## Additional files

### Supplementary files
MDAR checklist

### Data availability

Data and code for analysis is available at: https://doi.org/10.5281/zenodo.16271300. Source data is provided for figures where appropriate.

The following dataset was generated:

| Author(s) | Year | Dataset title | Dataset URL | Database and Identifier |
|---|---|---|---|---|
| Hosack VB, Arce-McShane FI | 2025 | OSMCx_3Ddirectionaltuning | https://doi.org/10.5281/zenodo.16271300 | Zenodo, 10.5281/zenodo.16271299 |

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

# Appendix 1

## Population sizes

**Appendix 1—table 1.** Numbers of individual neurons recorded from each array location during each data collection session.

| Control | rMIo | cMIo | SIo(3a/3b) | SIo(1/2) |
|---|---|---|---|---|
| R Feeding | 29 | 125 | 9 | 29 |
| Y Feeding | 28 | 76 | 18 | 37 |
| R Drinking | 31 | 185 | 23 | 54 |
| Y Drinking | 23 | 104 | 16 | 63 |
| Nerve block | | | | |
| R Feeding | 27 | 126 | 1 | 27 |
| Y Feeding | 29 | 64 | 17 | 21 |
| R Drinking | 36 | 182 | 26 | 56 |
| Y Drinking | 22 | 55 | 14 | 55 |

## Appendix 2

### Analysis of lingual yaw and pitch during feeding

To get a better understanding of directional tuning in three dimensions, we analyzed the lateral and vertical components of 3D tongue direction independently. The laterality of movements (yaw) was determined as the angle of rotation about the vertical axis ($\alpha$) and the vertical component (pitch) was the angle of rotation about the horizontal plane ($\varphi$):

$$\alpha = \tan^{-1}\left(\Delta z/\Delta x\right) \tag{1}$$

$$\varphi = \tan^{-1}\Delta y/\sqrt{\Delta x^2 + \Delta z^2} \tag{2}$$

where $x$, $y$, $z$ represent the position along the posterior-anterior, inferior-superior, and left-right axes, respectively, and $\Delta x, \Delta y, \Delta z$ represent the directional change in the position of the tongue tip over each 100 ms interval.

These results can be interpreted as the 3D direction of tongue movement broken down into its components. We made similar observations across the 3D angle, yaw, and pitch. This serves to reinforce our main results about the modulation of the OSMCx to tongue direction during feeding and how it is affected by the loss of oral tactile sensation.

There were similar distributions of yaw and pitch observed in both monkeys. For yaw, there was a multimodal distribution, with three peaks in the far right (positive), center, and far left. Conversely, there was a uniform distribution for pitch in the control condition, with the center near 0°, which became skewed toward the upward (positive) directions under the nerve block condition.

Many neurons exhibited modulation of spiking activity to the yaw and pitch angles of tongue movements. Consistent with our 3D angle results, fewer SIo neurons were directionally tuned to yaw and pitch compared to MIo. More neurons were tuned to the pitch direction than the yaw (chi-square, $p < 0.01$), except in the SIo of Monkey R where the difference was not significant. The changes in the proportion of directionally tuned neurons with the addition of nerve block were also comparable to the changes to 3D directionality. In most cases, there was a decrease with nerve block, except for the percentage of neurons tuned to yaw in Monkey Y, where there was an increase.

The distributions of preferred yaw and pitch were comparable between MIo and SIo in both subjects (circular k-test, $p > 0.3$), similar to the 3D angle. Overall, more neurons had an upward preferred pitch than a downward. Whereas in Monkey Y, there was a preference across areas for leftward yaw, in Monkey R the PDs were more evenly distributed. Following sensory loss, shifts in mean PD were observed in MIo and SIo of both animals for pitch and/or yaw (circular k-test, $p < 0.05$).

