## [Editor Report · eLife Assessment]

This study characterises motor and somatosensory cortex neural activity during naturalistic eating and drinking tongue movement in nonhuman primates. The data, which include electrophysiology, three-dimensional tracking of tongue movements, and nerve block manipulations, are **valuable** to neuroscientists and neural engineers interested in tongue use. Although the current analyses provide a **solid** description of single neuron activity in these areas, both the population level analyses and the characterisation of activity changes following nerve block could be improved.

---

## [Referee Report · Reviewer #1 (Public review)]

Summary:

Hosack and Arce-McShane investigate how the 3D movement direction of the tongue is represented in the orofacial part of the sensory-motor cortex and how this representation changes with the loss of oral sensation. They examine the firing patterns of neurons in the orofacial parts of the primary motor cortex (MIo) and somatosensory cortex (SIo) in non-human primates (NHPs) during drinking and feeding tasks. While recording neural activity, they also tracked the kinematics of tongue movement using biplanar video-radiography of markers implanted in the tongue. Their findings indicate that many units in both MIo and SIo are directionally tuned during the drinking task. However, during the feeding task, directional turning was more frequent in MIo units and less prominent in SIo units. Additionally, in some recording sessions, they blocked sensory feedback using bilateral nerve block injections, which seemed to result in fewer directionally tuned units and changes in the overall distribution of the preferred direction of the units.

Strengths:

The most significant strength of this paper lies in its unique combination of experimental tools. The author utilized a video-radiography method to capture 3D kinematics of the tongue movement during two behavioral tasks while simultaneously recording activity from two brain areas. This specific dataset and experimental setup hold great potential for future research on the understudied orofacial segment of the sensory-motor area.

Weaknesses:

A substantial portion of the paper is dedicated to establishing directional tuning in individual neurons, followed by an analysis of how this tuning changes when sensory feedback is blocked. While such characterizations are valuable, particularly in less-studied motor cortical areas and behaviors, the discrepancies in tuning changes across the two NHPs, coupled with the overall exploratory nature of the study, render the interpretation of these subtle differences somewhat speculative. At the population level, both decoding analyses and state space trajectories from factor analysis indicate that movement direction (or spout location) is robustly represented. However, as with the single-cell findings, the nuanced differences in neural trajectories across reach directions and between baseline and sensory-block conditions remain largely descriptive. To move beyond this, model-based or hypothesis-driven approaches are needed to uncover mechanistic links between neural state space dynamics and behavior.

---

## [Referee Report · Reviewer #2 (Public review)]

Summary:

This manuscript by Hosack and Arce-McShane examines the directional tuning of neurons in macaque primary motor (MIo) and somatosensory (SIo) cortex. The neural basis of tongue control is far less studied than, for example, forelimb movements, partly because the tongue's kinematics and kinetics are difficult to measure. A major technical advantage of this study is using biplanar video-radiography, processed with modern motion tracking analysis software, to track the movement of the tongue inside the oral cavity. Compared to prior work, the behaviors are more naturalistic behaviors (feeding and licking water from one of three spouts), although the animals were still head-fixed.

The study's main findings are that:

• A majority of neurons in MIo and a (somewhat smaller) percentage of SIo modulated their firing rates during tongue movements, with different modulation depending on the direction of movement (i.e., exhibited directional tuning). Examining the statistics of tuning across neurons, there was anisotropy (e.g., more neurons preferring anterior movement) and a lateral bias in which tongue direction neurons preferred that was consistent with the innervation patterns of tongue control muscles (although with some inconsistency between monkeys).

• Consistent with this encoding, tongue position could be decoded with moderate accuracy even from small ensembles of ~28 neurons.

• There were differences observed in the proportion and extent of directional tuning between the feeding and licking behaviors, with stronger tuning overall during licking. This potentially suggests behavioral context-dependent encoding.

• The authors then went one step further and used a bilateral nerve block to the sensory inputs (trigeminal nerve) from the tongue. This impaired the precision of tongue movements and resulted in an apparent reduction and change in neural tuning in MIo and SIo.

Strengths:

The data are difficult to obtain and appear to have been rigorously measured, and provide a valuable contribution to this under-explored subfield of sensorimotor neuroscience. The analyses adopt well-established methods especially from the arm motor control literature, and represent a natural starting point for characterizing tongue 3D direction tuning.

Weaknesses:

There are alternative explanations from some of the interpretations, but those interpretations are described in a way that clearly distinguishes results from interpretations, and readers can make their own assessments. Some of these limitations are described in more detail below.

One weakness of the current study is that there is substantial variability in some of the results between monkeys, including the tuning characteristics of primary somatosensory cortex neurons during drinking, and the effect of nerve block on tongue movements and the associated changes in single neuron tuning.

This study focuses on describing directional tuning using the preferred direction (PD) / cosine tuning model popularized by Georgopoulous and colleagues for understanding neural control of arm reaching in the 1980s. This is a reasonable starting point and a decent first order description of neural tuning. However, the arm motor control field has moved far past that viewpoint, and in some ways an over-fixation on static representational encoding models and PDs held that field back for many years. The manuscript benefit from drawing the readers' attention (perhaps in their Discussion) that PDs are a very simple starting point for characterizing how cortical activity relates to kinematics, but that there is likely much richer population-level dynamical structure and that a more mechanistic, control-focused analytical framework may be fruitful. A good review of this evolution in the arm field can be found in Vyas S, Golub MD, Sussillo D, Shenoy K. 2020. Computation Through Neural Population Dynamics. Annual Review of Neuroscience. 43(1):249-75. A revised version of the manuscript incorporates more population-level analyses, but with inconsistent use of quantifications/statistics and without sufficient contextualization of what the reader is to make of these results.

The described changes in tuning after nerve block could also be explained by changes in kinematics between these conditions, which temper the interpretation of these interesting results.

I am not convinced of the claim that tongue directional encoding fundamentally changes between drinking and feeding given the dramatically different kinematics and the involvement of other body parts like the jaw (e.g., the reference to Laurence-Chasen et al. 2023 just shows that there is tongue information independent of jaw kinematics, not that jaw movements don't affect these neurons' activities). I also find the nerve block results inconsistent (more tuning in one monkey, less in the other?) and difficult to really learn something fundamental from, besides that neural activity and behavior both change - in various ways - after nerve block (not at all surprising but still good to see measurements of).

The manuscript states that "Our results suggest that the somatosensory cortex may be less involved than the motor areas during feeding, possibly because it is a more ingrained and stereotyped behavior as opposed to tongue protrusion or drinking tasks". An alternative explanation be more statistical/technical in nature: that during feeding, there will be more variability in exactly what somatosensation afferent signals are being received from trial to trial (because slight differences in kinematics can have large differences in exactly where the tongue is and the where/when/how of what parts of it are touching other parts of the oral cavity)? This variability could "smear out" the apparent tuning using these types of trial-averaged analyses. Given how important proprioception and somatosensation are for not biting the tongue or choking, the speculation that somatosensory cortical activity is suppressed during feedback is very counter-intuitive to this reviewer. In the revised manuscript the authors note these potential confounds and other limitations in the Discussion.

---

## [Referee Report · Reviewer #3 (Public review)]

Summary

In this study, the authors aim to uncover how 3D tongue direction is represented in the Motor (M1o) and Somatosensory (S1o) cortex. In non-human primates implanted with chronic electrode arrays, they use X-ray based imaging to track the kinematics of the tongue and jaw as the animal is either chewing food or licking from a spout. They then correlate the tongue kinematics with the recorded neural activity. They perform both single-unit and population level analyses during feeding and licking. Then, they recharacterize the tuning properties after bilateral lidocaine injections in the two sensory branches of the trigeminal nerve. They report that their nerve block causes a reorganization of the tuning properties and population trajectories. Overall, this paper concludes that M1o and S1o both contain representations of the tongue direction, but their numbers, their tuning properties and susceptibility to perturbed sensory input are different.

Strengths

The major strengths of this paper are in the state-of-the-art experimental methods employed to collect the electrophysiological and kinematic data. In the revision, the single-unit analyses of tuning direction are robustly characterized. The differences in neural correlations across behaviors, regions and perturbations are robust. In addition to the substantial amount of largely descriptive analyses, this paper makes two convincing arguments (1) The single-neuron correlates for feeding and licking in OSMCx are different - and can't be simply explained by different kinematics and (2) Blocking sensory input alters the neural processing during orofacial behaviors. The evidence for these claims is solid.

Weaknesses

The main weakness of this paper is in providing an account for these differences to get some insight into neural mechanisms. For example, while the authors show changes in neural tuning and different 'neural trajectory' shapes during feeding and drinking - their analyses of these differences are descriptive and provide limited insight for the underlying neural computations.

---

## [Author Response]

The following is the authors’ response to the current reviews.

We have significant concerns about the eLife assessment and the reviews. The reviewers acknowledged substantial strengths in our work:

· Reviewer 3 noted that “the single-unit analyses of tuning direction are robustly characterized”, “the differences in neural correlations across behaviors, regions and perturbations are robust”, and “The evidence for these claims is solid

· Reviewer 2 stated that “the manuscript has been improved” with “new analyses [that] provide improved rigor”.

Despite these, the final eLife assessment inexplicably downplayed the significance of the findings and strength of evidence.

Broader Impact and Significance.

The findings, not only the data, have theoretical and/or practical implications extending well beyond a single subfield relevant to:

1. behavioral neuroscientists studying sensorimotor integration

2. systems and theoretical neuroscientists

3. neural and biomechanical engineers working on brain-computer interfaces for speech or oral or limb prosthetics

4. soft robotics researchers

5. comparative motor control researchers

6. clinicians involved in the evaluation and rehabilitation of orolingual function (e.g., after stroke or glossectomy, dysphagia)

Given this broad relevance, we question why the significance was characterized as merely “useful” rather than “important”.

Dismissive Tone Toward Descriptive Research.

Some reviews displayed a dismissive or skeptical tone of the findings and their significance, even when methods were solid and support for the claims were strong. They critiqued the “descriptive nature” of our study, faulting the lack of mechanistic explanation. However, in poorly understood fields such as orofacial sensorimotor control, descriptive studies provide the empirical foundation for mechanistic studies. Rich descriptive data generate testable hypotheses that drive mechanistic discoveries forward, while mechanistic studies conducted without this groundwork often pursue precise answers to poorly formulated questions.

Specific Issues with Reviews:

1. Significant omission in study description:

The eLife Assessment’s second sentence states: “The data, which include both electrophysiology and nerve block manipulations, will be of value to neuroscientists and neural engineers interested in tongue use.”

This description omits our simultaneously recorded high-resolution 3D kinematics data—a significant oversight given that combining high-density electrophysiological recording from multiple cortical regions with high-resolution 3D tongue kinematics during naturalistic behaviors in non-human primates represents one of our study's key strengths. Currently, only two research labs in the US possess this capability.

2. Overemphasis on the “smaller” and “inconsistent” findings

While we acknowledge some inconsistent findings between animals, the reviews overemphasized these inconsistencies in ways that cast unwarranted doubt on our more significant and consistent results.

a. Reviewer 1: “[...] the discrepancies in tuning changes across the two NHPs, coupled with the overall exploratory nature of the study, render the interpretation of these subtle differences somewhat speculative. “[...] in some recording sessions, they blocked sensory feedback using bilateral nerve block injections, which seemed to result in fewer directionally tuned units and changes in the overall distribution of the preferred direction of the units.”

The skeptical tone of the critique is in opposition to Reviewer 3’s statement that: “the evidence for these claims were solid”. In this statement, the reviewer characterized our findings as “somewhat speculative”, seemingly overlooking robust and consistent changes we documented

· “Following nerve block, MIo and SIo showed significant decreases in the proportion of directionally modulated neurons across both tasks (Fig. 10A; Chi-square, MIo: p <0.001, SIo: p < 0.05).”

· “Nerve block significantly altered PD distributions during both tasks. During feeding, MIo neurons in both subjects exhibited a significant clockwise shift in mean PD toward the center (0°), resulting in more uniform distributions (Fig. 11A; circular k-test, p < 0.01).”

· These results were obtained through careful subsampling of trials with similar kinematics for both feeding and drinking tasks, ensuring that the tuning changes in the nerve block experiments could not be attributed to differing kinematics.

b. Reviewer 2: “One weakness of the current study is that there is substantial variability in results between monkeys.”

This vague critique, without specifying which results showed “substantial variability”, reads as though most findings were inconsistent, unfairly casting doubt on our study’s validity.

3. Inaccurate statements in the Reviewers’ summaries

Several reviewer statements contain factual inaccuracies:

a. Reviewer 2: “A majority of neurons in MIo and a (somewhat smaller) percentage of SIo modulated their firing rates during tongue movements, with different modulation depending on the direction of movement (i.e., exhibited directional tuning).”

Reviewer 2's characterization of directional tuning misrepresents our findings. We reported substantial differences in the proportion of directionally tuned neurons between MIo and SIo during the feeding task but a smaller difference in the drinking task:

· “The proportion of directionally tuned neurons [...] differed significantly between MIo and SIo during the feeding task in both subjects (Chi-square, p < 0.001). In rostral and caudal MIo, 80% of neurons were modulated to 3D direction (bootstrap, p < 0.05, Fig. 3B, left), compared to 52% in areas 1/2 and 3a/3b.

· “During drinking, the proportion of directionally modulated neurons was more similar between regions (69% in MIo vs. 60% in SIo: Chi-square, p > 0.05, Fig. 3B right).”

b. Reviewer 2: “There were differences observed in the proportion and extent of directional tuning between the feeding and licking behaviors, with stronger tuning overall during licking.”

Reviewer 2's claim about task differences directly contradicts our findings. We consistently reported stronger tuning in feeding compared to drinking across multiple measures:

· “The proportion of directionally tuned neurons was higher in the feeding vs. drinking task (Chi-square, p < 0.05, feeding: 72%, drinking: 66%)”;

· “Cumulative explained variance for the first three factors was higher in feeding (MIo: 82%, SIo: 81%) than in drinking (MIo: 74%, SIo: 63%)”;

· “Decoding using LSTM showed consistently higher accuracies in feeding compared to drinking regardless of the length of intervals used ..., behavioral window .., and directional angles ...”

These results were also summarized in the Discussion.

c. Reviewer 1: In Figure 12, factor 2 and 3 are plotted against each other? and factor 1 is left out?

Reviewer 1’s observation about Figure 12 is incorrect. Factor 1 was included: Top subplots (feeding) show Factor 1 vs 3 (MIo) and Factor 1 vs 2 (SIo) while the bottom subplots (drinking) show Factor 2 vs 3 (MIo) and Factor 1 vs 2 (SIo). We plotted the two latent factors with highest explained variance for clarity, though all 20 factors were included in intertrajectory distance calculations.

4. Framing and interpretive over-scrutiny

Several critiques targeted framing rather than methodological rigor and emphasized that interpretations were speculative even when appropriately hedged:

a. Reviewer 2: “A revised version of the manuscript incorporates more population-level analyses, but with inconsistent use of quantifications/statistics and without sufficient contextualization of what the reader is to make of these results.”

Reviewer 2 mentioned “inconsistent use of quantifications/statistics” without specifying which analyses were problematic or updating their summary to include our additional population-level findings.

b. Reviewer 2: “The described changes in tuning after nerve block could also be explained by changes in kinematics between these conditions, which temper the interpretation of these interesting results”

Despite our addressing kinematic concerns through subsampled data analysis, Reviewer 2 remained unsatisfied, contrasting sharply with Reviewer 3's assessment that our arguments were “convincing” with “solid” evidence.

c. Reviewer 2: “I am not convinced of the claim that tongue directional encoding fundamentally changes between drinking and feeding given the dramatically different kinematics and the involvement of other body parts like the jaw

Reviewer 2 expressed skepticism about fundamental encoding differences between tasks, despite our comprehensive controls including subsampled data with similar kinematics and multiple verification analyses (equal neuron numbers, stable neurons, various interval lengths, behavioral windows, and directional angles).

Without describing why these analyses were insufficient, this criticism goes beyond methods or statistics. It casts doubt and challenges whether the conclusions are even worth drawing despite careful experimental controls.

d. Reviewer 2: “The manuscript states that “Our results suggest that the somatosensory cortex may be less involved than the motor areas during feeding, possibly because it is a more ingrained and stereotyped behavior as opposed to tongue protrusion or drinking tasks". Could an alternative explanation be more statistical/technical in nature: that during feeding, there will be more variability in exactly what somatosensation afferent signals are being received from trial to trial (because slight differences in kinematics can have large differences in exactly where the tongue is and the where/when/how of what parts of it are touching other parts of the oral cavity)? This variability could “smear out” the apparent tuning using these types of trial-averaged analyses. Given how important proprioception and somatosensation are for not biting the tongue or choking, the speculation that somatosensory cortical activity is suppressed during feedback is very counter-intuitive to this reviewer”

By not updating this section, Reviewer 2 failed to acknowledge our responsive revisions, including Fano factor analysis showing higher variability in SIo during feeding versus drinking, and our updated discussion addressing their concerns about trial-to-trial variability: “Varying tongue shape, tongue’s contact with varying bolus properties (size and texture) and other oral structures (palate, teeth) may weaken the directional signal contained in SIo activity. Thus, small differences in tongue kinematics might create large differences in sensory signals across trials. When looking at trial-averaged signals, this natural variability could make the neural response patterns appear less precise or specific than they are. These are consistent with our findings that for both tasks, spiking variability was higher in SIo.”

Authors’ Response to Recommendations for the authors:

We thank the editors and the reviewers for their helpful comments. We have provided a response to reviewers’ recommendations and made some revisions on the manuscript.

**Reviewer #1 (Recommendations for the authors):**
In the newly added population factor analysis, several methodological decisions remain unclear to me:In Figure 7, why do the authors compare the mean distance between conditions in the latent spaces of MIo and SIo? Since these latent spaces are derived separately, they exist on different scales (with MIo appearing roughly four times larger than SIo), and this discrepancy is reflected in the reported mean distances (Figure 7, inset plots). Wouldn't this undermine a direct comparison?

Thank you for this helpful feedback. The reviewer is correct that the latent spaces are derived separately for MIo and SIo, thus they exist on different scales as we have noted in the caption of Figure 7: “Axes for SIo are 1/4 scale of MIo.”

To allow for a direct comparison between MIo and SIo, we corrected the analysis by comparing their normalized mean inter-trajectory distances obtained by first calculating the geometric index (*GI*) of the inter-trajectory distances, *d*, between each pair of population trajectories per region as: GI = (d_1_-d_2_)/ (d_1_+d_2_). We then performed the statistics on the GIs and found a significant difference between mean inter-trajectory distances in MIo vs. SIo. We performed the same analysis comparing the distance travelled between MIo and SIo trajectories by getting the normalized difference in distances travelled and still found a significant difference in both tasks. We have updated the results and figure inset to reflect these changes.

In Figure 12, unlike Figure 7 which shows three latent dimensions, only two factors are plotted. While the methods section describes a procedure for selecting the optimal number of latent factors, Figure 7 - figure supplement 3 shows that variance explained continues to increase up to about five latent dimensions across all areas. Why, then, are fewer dimensions shown?

Thank you for the opportunity to clarify the figure. The ***m*** obtained from the 3-fold crossvalidation varied for the full sample and was 20 factors for the subsample. We clarify that all statistical analyses were done using 20 latent factors. Using the full sample of neurons, the first 3 factors explained 81% of variance in feeding data compared to 71% in drinking data. When extended to 5 factors, feeding maintained its advantage with 91% variance explained versus 82% for drinking. Because feeding showed higher variance explained than drinking across 3 or 5 factors, only three factors were shown in Figure 7 for better visualization. We added this clarification to the Methods and Results.

Figure 12 shows the differences in the neural trajectories between the control and nerve block conditions. The control vs. nerve block comparison complicated the visualization of the results. Thus, we plotted only the two latent factors with the highest separation between population trajectories. This was clarified in the Methods and caption of Figure 12.

In Figure 12, factor 2 and 3 are plotted against each other? and factor 1 is left out?

This observation is incorrect; Factor 1 was included: Top subplots (feeding) show Factor 1 vs 3 (MIo) and Factor 1 vs 2 (SIo) while the bottom subplots (drinking) show Factor 2 vs 3 (MIo) and Factor 1 vs 2 (SIo). We have clarified this in the Methods and caption of Figure 12.

Finally, why are factor analysis results shown only for monkey R?

Factor analysis results were performed on both animals, but the results were shown only for Monkey R to decrease the number of figures in the manuscript. Figure 7- figure supplement 1 shows the data for both monkeys. Here are the equivalent Figure 7 plots for monkey Y.

**Author response image 1. sa4fig1:** 

**Reviewer #2 (Recommendations for the authors):**
Overall, the manuscript has been improved.New analyses provide improved rigor (as just one example, organizing the feeding data into three-category split to better match the three-direction drinking data decoding analysis and also matching the neuron counts).The updated nerve block change method (using an equal number of trials with a similar leftright angle of movement in the last 100 ms of the tongue trajectory) somewhat reduces my concern that kinematic differences could account for the neural changes, but on the other hand the neural analyses use 250 ms (meaning that the neural differences could be related to behavioral differences earlier in the trial). Why not subselect to trials with similar trajectories throughout the whole movement(or at least show that as an additional analysis, albeit one with lower trial counts).

As the reviewer pointed out, selecting similar trajectories throughout the whole movement would result in lower trial counts that lead to poor statistical power. We think that the 100 ms prior to maximum tongue protrusion is a more important movement segment to control for similar kinematics between the control and nerve block conditions since this represents the subject’s intended movement endpoint.

A lot of the Results seemed like a list of measurements without sufficient hand-holding or guide-posting to explain what the take-away for the reader should be. Just one example to make concrete this broadly-applicable feedback: "Cumulative explained variance for the first three factors was higher in feeding (MIo: 82%, SIo: 81%) than in drinking (MIo: 74%, SIo: 63%) when all neurons were used for the factor analysis (Fig. 7)": why should we care about 3 factors specifically? Does this mean that in feeding, the neural dimensionality is lower (since 3 factors explain more of it)? Does that mean feeding is a "simpler" behavior (which is counter-intuitive and does not conform to the authors' comments about the higher complexity of feeding). And from later in that paragraph: what are we do make of the differences in neural trajectory distances (aside from quantifying using a different metric the same larger changes in firing rates that could just as well be quantified as statistics across single-neuron PETHs)?

Thank you for the feedback on the writing style. We have made some revisions to describe the takeaway for the reader. That fewer latent factors explain 80% of the variance in the feeding data means that the underlying network activity is relatively simple despite apparent complexity. When neural population trajectories are farther away from each other in state space, it means that the patterns of activity across tongue directions are more distinct and separable, thus, less likely to be confused with each other. This signifies that neural representations of 3D tongue directions are more robust. When there is better neural discrimination and more reliable information processing, it is easier for downstream brain regions to distinguish between different tongue directions.

The addition of more population-level analyses is nice as it provides a more efficient summary of the neural measurements. However, it's a surface-level dive into these methods; ultimately the goal of ensemble "computation through dynamics" analyses is to discover simpler structure / organizational principles at the ensemble level (i.e., show things not evidence from single neurons), rather than just using them as a way to summarize data. For instance, here neural rotations are remarked upon in the Results, without referencing influential prior work describing such rotations and why neural circuits may use this computational motif to separate out conditions and shape muscle activity-generating readouts (Churchland et al. Nature 2012 and subsequent theoretical iterations including the Russo et al.). That said, the Russo et al tangling study was well-referenced and the present tangling results were effectively contextualized with respect to that paper in terms of the interpretation. I wish more of the results were interpreted with comparable depth.Speaking of Russo et al: the authors note qualitative differences in tangling between brain areas, but do not actually quantify tangling in either. These observations would be stronger if quantified and accompanied with statistics.

Contrary to the reviewer’s critique, we did frame these results in the context of structure/organizational principles at the ensemble level. We had already cited prior work of Churchland et al., 2012; Michaels et al., 2016and Russo et al., 2018. In the Discussion, Differences across behaviors, we wrote: “In contrast, MIo trajectories in drinking exhibited a consistent rotational direction regardless of spout location (Fig. 7). This may reflect a predominant non-directional information such as condition-independent time-varying spiking activity during drinking (Kaufman et al., 2016; Kobak et al., 2016; Arce-McShane et al., 2023).”

Minor suggestions:Some typos, e.g.• no opening parenthesis in "We quantified directional differences in population activity by calculating the Euclidean distance over m latent factors"• missing space in "independent neurons(Santhanam et al., 2009;...");• missing closing parentheses in "followed by the Posterior Inferior (Figure 3 - figure supplement 1)."There is a one-page long paragraph in the Discussion. Please consider breaking up the text into more paragraphs each organized around one key idea to aid readability.

Thank you, we have corrected these typos.

Could it be that the Kaufman et al 2013 reference was intended to be Kaufman et al 2015 eNeuro (the condition-invariant signal paper)?

Thank you, we have corrected this reference.

At the end of the Clinical Implications subsection of the Discussion, the authors note the growing field of brain-computer interfaces with references for motor read-out or sensory write-in of hand motor/sensory cortices, respectively. Given that this study looks at orofacial cortices, an even more clinically relevant development is the more recent progress in speech BCIs (two recent reviews: https://www.nature.com/articles/s41583-024-00819-9, https://www.annualreviews.org/content/journals/10.1146/annurev-bioeng-110122012818) many of which record from human ventral motor cortex and aspirations towards FES-like approaches for orofacial movements (e.g., https://link.springer.com/article/10.1186/s12984-023-01272-y).

Thank you, we have included these references.

**Reviewer #3 (Recommendations for the authors):**
Major Suggestions(1) For the factor analysis of feeding vs licking, it appears that the factors were calculated separately for the two behaviors. It could be informative to calculate the factors under both conditions and project the neural data for the two behaviors into that space. The overlap/separations of the subspace could be informative.

We clarify that we performed a factor analysis that included both feeding and licking for MIo, as stated in the Results: “To control for factors such as different neurons and kinematics that might influence the results, we performed factor analysis on stable neurons across both tasks using all trials (Fig. 7- figure supplement 2A) and using trials with similar kinematics (Fig. 7- figure supplement 2B).” We have revised the manuscript to reflect this more clearly.

(2) For the LSTM, the Factor analyses and the decoding it is unclear if the firing rates are mean subtracted and being normalized (the methods section was a little unclear). Typically, papers in the field either z-score the data or do a softmax.

The firing rates were z-scored for the LSTM and KNN. For the factor analysis, the spike counts were not z-scored, but the results were normalized. We clarified this in the Methods section.

Minor:Page 1: Abstract- '... how OSMCx contributes to...'Since there are no direct causal manipulations of OSMCx in this manuscript, this study doesn't directly study the OSMCx's contribution to movement - I would recommend rewording this sentence.Similarly, Page 2: 'OSMCx plays an important role in coordination...' the citations in this paragraph are correlative, and do not demonstrate a causal role.There are similar usages of 'OSMCx coordinates...' in other places e.g. Page 8.

Thank you, we revised these sentences.

Page 7: the LSTM here has 400 units, which is a very large network and contains >12000 parameters. Networks of this size are prone to memorization, it would be wise to test the rsquare of the validation set against a shuGled dataset to see if the network is actually working as intended.

Thank you for bringing up this important point of verifying that the network is learning meaningful patterns versus memorizing. Considering the size of our training samples, the ratio of samples to parameters is appropriate and thus the risk of memorization is low. Indeed, validation tests and cross-validation performed indicated expected network behavior and the R squared values obtained here were similar to those reported in our previous paper (Laurence-Chasen et al., 2023).

The following is the authors’ response to the original reviews

**Public Reviews:**

**Reviewer #1 (Public review):**
Summary:In their paper, Hosack and Arce-McShane investigate how the 3D movement direction of the tongue is represented in the orofacial part of the sensory-motor cortex and how this representation changes with the loss of oral sensation. They examine the firing patterns of neurons in the orofacial parts of the primary motor cortex (MIo) and somatosensory cortex (SIo) in non-human primates (NHPs) during drinking and feeding tasks. While recording neural activity, they also tracked the kinematics of tongue movement using biplanar videoradiography of markers implanted in the tongue. Their findings indicate that most units in both MIo and SIo are directionally tuned during the drinking task. However, during the feeding task, directional turning was more frequent in MIo units and less prominent in SIo units. Additionally, in some recording sessions, they blocked sensory feedback using bilateral nerve block injections, which resulted in fewer directionally tuned units and changes in the overall distribution of the preferred direction of the units.Strengths:The most significant strength of this paper lies in its unique combination of experimental tools. The author utilized a video-radiography method to capture 3D kinematics of the tongue movement during two behavioral tasks while simultaneously recording activity from two brain areas. Moreover, they employed a nerve-blocking procedure to halt sensory feedback. This specific dataset and experimental setup hold great potential for future research on the understudied orofacial segment of the sensory-motor area.Weaknesses:Aside from the last part of the result section, the majority of the analyses in this paper are focused on single units. I understand the need to characterize the number of single units that directly code for external variables like movement direction, especially for less-studied areas like the orofacial part of the sensory-motor cortex. However, as a field, our decadelong experience in the arm region of sensory-motor cortices suggests that many of the idiosyncratic behaviors of single units can be better understood when the neural activity is studied at the level of the state space of the population. By doing so, for the arm region, we were able to explain why units have "mixed selectivity" for external variables, why the tuning of units changes in the planning and execution phase of the movement, why activity in the planning phase does not lead to undesired muscle activity, etc. See (Gallego et al. 2017; Vyas et al. 2020; Churchland and Shenoy 2024) for a review. Therefore, I believe investigating the dynamics of the population activity in orofacial regions can similarly help the reader go beyond the peculiarities of single units and in a broader view, inform us if the same principles found in the arm region can be generalized to other segments of sensorymotor cortex.

We thank and agree with the reviewer on the value of information gained from studying population activity. We also appreciate that population analyses have led to the understanding that individual neurons have “mixed selectivity”. We have shown previously that OSMCx neurons exhibit mixed selectivity in their population activity and clear separation between latent factors associated with gape and bite force levels (Arce-McShane FI, Sessle BJ, Ram Y, Ross CF, Hatsopoulos NG (2023) Multiple regions of primate orofacial sensorimotor cortex encode bite force and gape. Front Systems Neurosci. doi: 10.3389/fnsys.2023.1213279. PMID: 37808467 PMCID: 10556252), and chew-side and food types (Li Z & Arce-McShane FI (2023). Cortical representation of mastication in the primate orofacial sensorimotor cortex. Program No. NANO06.05. 2023 Neuroscience Meeting Planner. Washington, D.C.: Society for Neuroscience, 2023. Online.).

The primary goal of this paper was to characterize single units in the orofacial region and to do a follow-up paper on population activity. In the revised manuscript, we have now incorporated the results of population-level analyses. The combined results of the single unit and population analyses provide a deeper understanding of the cortical representation of 3D direction of tongue movements during natural feeding and drinking behaviors.

Further, for the nerve-blocking experiments, the authors demonstrate that the lack of sensory feedback severely alters how the movement is executed at the level of behavior and neural activity. However, I had a hard time interpreting these results since any change in neural activity after blocking the orofacial nerves could be due to either the lack of the sensory signal or, as the authors suggest, due to the NHPs executing a different movement to compensate for the lack of sensory information or the combination of both of these factors. Hence, it would be helpful to know if the authors have any hint in the data that can tease apart these factors. For example, analyzing a subset of nerve-blocked trials that have similar kinematics to the control.

Thank you for bringing this important point. We agree with the reviewer that any change in the neural activity may be attributed to lack of sensory signal or to compensatory changes or a combination of these factors. To tease apart these factors, we sampled an equal number of trials with similar kinematics for both control and nerve block feeding sessions. We added clarifying description of this approach in the Results section of the revised manuscript: “To confirm this effect was not merely due to altered kinematics, we conducted parallel analyses using carefully subsampled trials with matched kinematic profiles from both control and nerve-blocked conditions.”

Furthermore, we ran additional analysis for the drinking datasets by subsampling a similar distribution of drinking movements from each condition. We compared the neural data from an equal number of trials with a similar left-right angle of movement in the last 100 ms of the tongue trajectory, nearest the spout. We compared the directional tuning across an equal number of trials with a similar left-right angle of movement in the last 100 ms of the tongue trajectory, nearest the spout. These analyses that control for similar kinematics showed that there was still a decrease in the proportion of directionally modulated neurons with nerve block compared to the control. This confirms that the results may be attributed to the lack of tactile information. These are now integrated in the revised paper under Methods section: Directional tuning of single neurons, as well as Results section: Effects of nerve block: Decreased directional tuning of MIo and SIo neurons and Figure 10 – figure supplement 1.

**Reviewer #2 (Public review):**
Summary:This manuscript by Hosack and Arce-McShane examines the directional tuning of neurons in macaque primary motor (MIo) and somatosensory (SIo) cortex. The neural basis of tongue control is far less studied than, for example, forelimb movements, partly because the tongue's kinematics and kinetics are difficult to measure. A major technical advantage of this study is using biplanar video-radiography, processed with modern motion tracking analysis software, to track the movement of the tongue inside the oral cavity. Compared to prior work, the behaviors are more naturalistic behaviors (feeding and licking water from one of three spouts), although the animals were still head-fixed.The study's main findings are that:• A majority of neurons in MIo and a (somewhat smaller) percentage of SIo modulated their firing rates during tongue movements, with different modulations depending on the direction of movement (i.e., exhibited directional tuning). Examining the statistics of tuning across neurons, there was anisotropy (e.g., more neurons preferring anterior movement) and a lateral bias in which tongue direction neurons preferred that was consistent with the innervation patterns of tongue control muscles (although with some inconsistency between monkeys).• Consistent with this encoding, tongue position could be decoded with moderate accuracy even from small ensembles of ~28 neurons.• There were differences observed in the proportion and extent of directional tuning between the feeding and licking behaviors, with stronger tuning overall during licking. This potentially suggests behavioral context-dependent encoding.• The authors then went one step further and used a bilateral nerve block to the sensory inputs (trigeminal nerve) from the tongue. This impaired the precision of tongue movements and resulted in an apparent reduction and change in neural tuning in Mio and SIo.Strengths:The data are difficult to obtain and appear to have been rigorously measured, and provide a valuable contribution to this under-explored subfield of sensorimotor neuroscience. The analyses adopt well-established methods, especially from the arm motor control literature, and represent a natural starting point for characterizing tongue 3D direction tuning.Weaknesses:There are alternative explanations for some of the interpretations, but those interpretations are described in a way that clearly distinguishes results from interpretations, and readers can make their own assessments. Some of these limitations are described in more detail below.One weakness of the current study is that there is substantial variability in results between monkeys, and that only one session of data per monkey/condition is analyzed (8 sessions total). This raises the concern that the results could be idiosyncratic. The Methods mention that other datasets were collected, but not analyzed because the imaging pre-processing is very labor-intensive. While I recognize that time is precious, I do think in this case the manuscript would be substantially strengthened by showing that the results are similar on other sessions.

We acknowledge the reviewer’s concern about inter-subject variability. Animal feeding and drinking behaviors are quite stable across sessions, thus, we do not think that additional sessions will address the concern that the results could be idiosyncratic. Each of the eight datasets analyzed here have su icient neural and kinematic data to capture neural and behavioral patterns. Nevertheless, we performed some of the analyses on a second feeding dataset from Monkey R. The results from analyses on a subset of this data were consistent across datasets; for example, (1) similar proportions of directionally tuned neurons, (2) similar distances between population trajectories (t-test p > 0.9), and (3) a consistently smaller distance between Anterior-Posterior pairs than others in MIo (t-test p < 0.05) but not SIo (p > 0.1).

This study focuses on describing directional tuning using the preferred direction (PD) / cosine tuning model popularized by Georgopoulous and colleagues for understanding neural control of arm reaching in the 1980s. This is a reasonable starting point and a decent first-order description of neural tuning. However, the arm motor control field has moved far past that viewpoint, and in some ways, an over-fixation on static representational encoding models and PDs held that field back for many years. The manuscript benefits from drawing the readers' attention (perhaps in their Discussion) that PDs are a very simple starting point for characterizing how cortical activity relates to kinematics, but that there is likely much richer population-level dynamical structure and that a more mechanistic, control-focused analytical framework may be fruitful. A good review of this evolution in the arm field can be found in Vyas S, Golub MD, Sussillo D, Shenoy K. 2020. Computation Through Neural Population Dynamics. Annual Review of Neuroscience. 43(1):249-75

Thank you for highlighting this important point. Research on orofacial movements hasn't progressed at the same pace as limb movement studies. Our manuscript focused specifically on characterizing the 3D directional tuning properties of individual neurons in the orofacial area—an analysis that has not been conducted previously for orofacial sensorimotor control. While we initially prioritized this individual neuron analysis, we recognize the value of broader population-level insights.

Based on your helpful feedback, we have incorporated additional population analyses to provide a more comprehensive picture of orofacial sensorimotor control and expanded our discussion section. We appreciate your expertise in pushing our work to be more thorough and aligned with current neuroscience approaches.

Can the authors explain (or at least speculate) why there was such a large difference in behavioral effect due to nerve block between the two monkeys (Figure 7)?

We acknowledge this as a variable inherent to this type of experimentation. Previous studies have found large kinematic variation in the effect of oral nerve block as well as in the following compensatory strategies between subjects. Each animal’s biology and response to perturbation vary naturally. Indeed, our subjects exhibited different feeding behavior even in the absence of nerve block perturbation (see Figure 2 in Laurence-Chasen et al., 2022). This is why each individual serves as its own control.

Do the analyses showing a decrease in tuning after nerve block take into account the changes (and sometimes reduction in variability) of the kinematics between these conditions? In other words, if you subsampled trials to have similar distributions of kinematics between Control and Block conditions, does the effect hold true? The extreme scenario to illustrate my concern is that if Block conditions resulted in all identical movements (which of course they don't), the tuning analysis would find no tuned neurons. The lack of change in decoding accuracy is another yellow flag that there may be a methodological explanation for the decreased tuning result.

Thank you for bringing up this point. We accounted for the changes in the variability of the kinematics between the control and nerve block conditions in the feeding dataset where we sampled an equal number of trials with similar kinematics for both control and nerve block. However, we did not control for similar kinematics in the drinking task. In the revised manuscript, we have clarified this and performed similar analysis for the drinking task. We sampled a similar distribution of drinking movements from each condition. We compared the neural data from an equal number of trials with a similar left-right angle of movement in the last 100 ms of the tongue trajectory, nearest the spout. There was a decrease in the percentage of neurons that were directionally modulated (between 30 and 80%) with nerve block compared to the control. These results have been included in the revised paper under Methods section: Directional tuning of single neurons, as well as Results section: Effects of nerve block: Decreased directionality of MIo and SIo neurons.

While the results from decoding using KNN did not show significant differences between decoding accuracies in control vs. nerve block conditions, the results from the additional factor analysis and decoding using LSTM were consistent with the decrease in directional tuning at the level of individual neurons.

The manuscript states that "Our results suggest that the somatosensory cortex may be less involved than the motor areas during feeding, possibly because it is a more ingrained and stereotyped behavior as opposed to tongue protrusion or drinking tasks". Could an alternative explanation be more statistical/technical in nature: that during feeding, there will be more variability in exactly what somato sensation afferent signals are being received from trial to trial (because slight differences in kinematics can have large differences in exactly where the tongue is and the where/when/how of what parts of it are touching other parts of the oral cavity)? This variability could "smear out" the apparent tuning using these types of trial-averaged analyses. Given how important proprioception and somatosensation are for not biting the tongue or choking, the speculation that somatosensory cortical activity is suppressed during feedback is very counter-intuitive to this reviewer.

Thank you for bringing up this point. We have now incorporated this in our revised Discussion (see Comparison between MIo and SIo). We agree with the reviewer that trialby-trial variability in the a erent signals may account for the lower directional signal in SIo during feeding than in drinking. Indeed, SIo’s mean-matched Fano factor in feeding was significantly higher than those in drinking (Author response image 1). Moreover, the results of the additional population and decoding analyses also support this.

**Author response image 2. sa4fig2:** Comparison of mean-matched Fano Factor between Sio neurons during feeding and drinking control tasks across both subjects (Wilcoxon rank sum test, p < 0. 001).

**Reviewer #3 (Public review):**
Summary:In this study, the authors aim to uncover how 3D tongue direction is represented in the Motor (M1o) and Somatosensory (S1o) cortex. In non-human primates implanted with chronic electrode arrays, they use X-ray-based imaging to track the kinematics of the tongue and jaw as the animal is either chewing food or licking from a spout. They then correlate the tongue kinematics with the recorded neural activity. Using linear regressions, they characterize the tuning properties and distributions of the recorded population during feeding and licking. Then, they recharacterize the tuning properties after bilateral lidocaine injections in the two sensory branches of the trigeminal nerve. They report that their nerve block causes a reorganization of the tuning properties. Overall, this paper concludes that M1o and S1o both contain representations of the tongue direction, but their numbers, their tuning properties, and susceptibility to perturbed sensory input are different.Strengths:The major strengths of this paper are in the state-of-the-art experimental methods employed to collect the electrophysiological and kinematic data.Weaknesses:However, this paper has a number of weaknesses in the analysis of this data.It is unclear how reliable the neural responses are to the stimuli. The trial-by-trial variability of the neural firing rates is not reported. Thus, it is unclear if the methods used for establishing that a neuron is modulated and tuned to a direction are susceptible to spurious correlations. The authors do not use shuffling or bootstrapping tests to determine the robustness of their fits or determining the 'preferred direction' of the neurons. This weakness colors the rest of the paper.Thank you for raising these points. We have performed the following additional analyses: (1) We have added analyses to ensure that the results could not be explained by neural variability. To show the trial-by-trial variability of the neural firing rates, we have calculated the Fano factor (mean overall = 1.34747; control = 1.46471; nerve block = 1.23023). The distribution was similar across directions, suggesting that responses of MIo and SIo neurons to varying 3D directions were reliable. (2) We have used a bootstrap procedure to ensure that directional tuning cannot be explained by mere chance. (3) To test the robustness of our PDs we also performed a bootstrap test, which yielded the same results for >90% of neurons, and a multiple linear regression test for fit to a cosine-tuning function. In the revised manuscript, the Methods and Results sections have been updated to include these analyses.

**Author response image 3. sa4fig3:** Comparison of Fano Factor across directions for MIo and SIo Feeding Control (Kruskal-Wallis, p > 0. 7).

The authors compare the tuning properties during feeding to those during licking but only focus on the tongue-tip. However, the two behaviors are different also in their engagement of the jaw muscles. Thus many of the differences observed between the two 'tasks' might have very little to do with an alternation in the properties of the neural code - and more to do with the differences in the movements involved.

Using the tongue tip for the kinematic analysis of tongue directional movements was a deliberate choice as the anterior region of the tongue is highly mobile and sensitive due to a higher density of mechanoreceptors. The tongue tip is the first region that touches the spout in the drinking task and moves the food into the oral cavity for chewing and subsequent swallowing.

We agree with the reviewer that the jaw muscles are engaged differently in feeding vs. drinking (see Fig. 2). For example, a wider variety of jaw movements along the three axes are observed in feeding compared to the smaller amplitude and mostly vertical jaw movements in drinking. Also, the tongue movements are very different between the two behaviors. In feeding, the tongue moves in varied directions to position the food between left-right tooth rows during chewing, whereas in the drinking task, the tongue moves to discrete locations to receive the juice reward. Moreover, the tongue-jaw coordination differs between tasks; maximum tongue protrusion coincides with maximum gape in drinking but with minimum gape in the feeding behavior. Thus, the different tongue and jaw movements required in each behavior may account for some of the differences observed in the directional tuning properties of individual neurons and population activity. These points have been included in the revised Discussion.

**Author response image 4. sa4fig4:** Tongue tip position (mm) and jaw pitch(degree) during feeding (left) and drinking (right) behaviors. Most protruded tongue position coincides with minimum gape (jaw pitch at 0°) during feeding but with maximum gape during drinking.

Many of the neurons are likely correlated with both Jaw movements and tongue movements - this complicates the interpretations and raises the possibility that the differences in tuning properties across tasks are trivial.

We thank the reviewer for raising this important point. In fact, we verified in a previous study whether the correlation between the tongue and jaw kinematics might explain differences in the encoding of tongue kinematics and shape in MIo (see Supplementary Fig. 4 in Laurence-Chasen et al., 2023): “Through iterative sampling of sub-regions of the test trials, we found that correlation of tongue kinematic variables with mandibular motion does not account for decoding accuracy. Even at times where tongue motion was completely un-correlated with the jaw, decoding accuracy could be quite high.”

The results obtained from population analyses showing distinct properties of population trajectories in feeding vs. drinking behaviors provide strong support to the interpretation that directional information varies between these behaviors.

The population analyses for decoding are rudimentary and provide very coarse estimates (left, center, or right), it is also unclear what the major takeaways from the population decoding analyses are. The reduced classification accuracy could very well be a consequence of linear models being unable to account for the complexity of feeding movements, while the licking movements are 'simpler' and thus are better accounted for.

We thank the reviewer for raising this point. The population decoding analyses provide additional insight on the directional information in population activity, as well as a point of comparison with the results of numerous decoding studies on the arm region of the sensorimotor cortex. In the revised version, we have included the results from decoding tongue direction using a long short-term memory (LSTM) network for sequence-tosequence decoding. These results differed from the KNN results, indicating that a linear model such as KNN was better for drinking and that a non-linear and continuous decoder was better suited for feeding. These results have been included in the revised manuscript.

The nature of the nerve block and what sensory pathways are being affected is unclear - the trigeminal nerve contains many different sensory afferents - is there a characterization of how effectively the nerve impulses are being blocked? Have the authors confirmed or characterized the strength of their inactivation or block, I was unable to find any electrophysiological evidence characterizing the perturbation.

The strength of the nerve block is characterized by a decrease in the baseline firing rate of SIo neurons, as shown in Supplementary Figure 6 of “Loss of oral sensation impairs feeding performance and consistency of tongue–jaw coordination” (Laurence-Chasen et al., 2022)..

Overall, while this paper provides a descriptive account of the observed neural correlations and their alteration by perturbation, a synthesis of the observed changes and some insight into neural processing of tongue kinematics would strengthen this paper.

We thank the reviewer for this suggestion. We have revised the Discussion to provide a synthesis of the results and insights into the neural processing of tongue kinematics.

**Recommendations for the authors:**

**Reviewer #1 (Recommendations for the authors):**
(1) The procedure for anesthesia explained in the method section was not clear to me. The following information was missing: what drug/dose was used? How long the animal was under anesthesia? How long after the recovery the experiments were done?

The animals were fully sedated with ketamine (100 mg/ml, 10 mg/kg) for less than 30 minutes, and all of the data was collected within 90 minutes after the nerve block was administered.

(2) In Figure 10, panels A and B are very close together, it was not at first clear whether the text "Monkey R, Monkey Y" belongs to panel A or B.

We have separated the two panels further in the revised figure.

(3) I found Figure 11 very busy and hard to interpret. Separating monkeys, fitting the line for each condition, or using a bar plot can help with the readability of the figure.

Thank you for the suggestion. We agree with you and have reworked this figure. To simplify it we have shown the mean accuracy across iterations.

(4) I found the laterality discussions like "This signifies that there are more neurons in the left hemisphere contributes toward one direction of tongue movement, suggesting that there is some laterality in the PDs of OSMCx neurons that varies between individuals" bit of an over-interpretation of data, given the low n value and the dissimilarity in how strongly the nerve blocking altered monkies behavior.

Thank you for sharing this viewpoint. We do think that laterality is a good point of comparison with studies on M1 neurons in the arm/hand region. In our study, we found that the peak of the PD distribution coincides with leftward tongue movements in feeding. The distribution of PDs provides insight into how tongue muscles are coordinated during movement. Intrinsic and extrinsic tongue muscles are involved in shaping the tongue (e.g., elongation, broadening) and positioning the tongue (e.g., protrusion/retraction, elevation/depression), respectively. These muscles receive bilateral motor innervation except for genioglossus. Straight tongue protrusion requires the balanced action of the right and left genioglossi while the lateral protrusion involves primarily the contralateral genioglossus. Given this unilateral innervation pattern, we hypothesized that left MIo/SIo neurons would preferentially respond to leftward tongue movements, corresponding to right genioglossus activation.

**Reviewer #2 (Recommendations for the authors):**
Are the observation of tuning peaks being most frequently observed toward the anterior and superior directions consistent with the statistics of the movements the tongue typically makes? This could be analogous to anisotropies previously reported in the arm literature, e.g., Lillicrap TP, Scott SH. 2013. Preference Distributions of Primary Motor Cortex Neurons Reflect Control Solutions Optimized for Limb Biomechanics. Neuron. 77(1):168-79

Thank you for bringing our attention to analogous findings by Lillicrap & Scott, 2013. Indeed, we do observe the highest number of movements in the Anterior Superior directions, followed by the Posterior Inferior. This does align with the distribution of tuning peaks that we observed. Author response image 4 shows the proportions of observed movements in each group of directions across all feeding datasets. We have incorporated this data in the Results section: Neuronal modulation patterns differ between MIo and SIo, as well as added this point in the Discussion.

**Author response image 5. sa4fig5:** Proportion of feeding trials in each group of directions. Error bars represent ±1 standard deviation across datasets (n = 4).

"The Euclidean distance was used to identify nearest neighbors, and the number of nearest neighbors used was K = 7. This K value was determined after testing different Ks which yielded comparable results." In general, it's a decoding best practice to tune hyperparameters (like K) on fully held-out data from the data used for evaluation. Otherwise, this tends to slightly inflate performance because one picks the hyperparameter that happened to give the best result. It sounds like that held-out validation set wasn't used here. I don't think that's going to change the results much at all (especially given the "comparable results" comment), but providing this suggestion for the future. If the authors replicate results on other datasets, I suggest they keep K = 7 to lock in the method.

K = 7 was chosen based on the size of our smallest training dataset (n = 55). The purpose of testing different K values was not to select which value gave the best result, but to demonstrate that similar K values did not affect the results significantly. We tested the different K values on a subset of the feeding data, but that data was not fully held-out from the training set. We will keep your suggestion in mind for future analysis.

The smoothing applied to Figure 2 PSTHs appears perhaps excessive (i.e., it may be obscuring interesting finer-grained details of these fast movements). Can the authors reduce the 50 ms Gaussian smoothing (I assume this is the s.d.?) ~25 ms is often used in studying arm kinematics. It also looks like the movement-related modulation may not be finished in these 200 ms / 500 ms windows. I suggest extending the shown time window. It would also be helpful to show some trial-averaged behavior (e.g. speed or % displacement from start) under or behind the PSTHs, to give a sense of what phase of the movement the neural activity corresponds to.

Thank you for the suggestion. We have taken your suggestions into consideration and modified Figure 2 accordingly. We decreased the Gaussian kernel to 25 ms and extended the time window shown. The trial-averaged anterior/posterior displacement was also added to the drinking PSTHs.

**Reviewer #3 (Recommendations for the authors):**
The major consideration here is that the data reported for feeding appears to be very similar to that reported in a previous study:"Robust cortical encoding of 3D tongue shape during feeding in macaques"Are the neurons reported here the same as the ones used in this previous paper? It is deeply concerning that this is not reported anywhere in the methods section.

These are the same neurons as in our previous paper, though here we include several additional datasets of the nerve block and drinking sessions. We have now included this in the methods section.

Second, I strongly recommend that the authors consider a thorough rewrite of this manuscript and improve the presentation of the figures. As written, it was not easy to follow the paper, the logic of the experiments, or the specific data being presented in the figures.

Thank you for this suggestion. We have done an extensive rewrite of the manuscript and revision of the figures.

A few recommendations:(1) Please structure your results sections and use descriptive topic sentences to focus the reader. In the current version, it is unclear what the major point being conveyed for each analysis is.

Thank you for this suggestion. We have added topic sentences to the begin each section of the results.

(2) Please show raster plots for at least a few example neurons so that the readers have a sense of what the neural responses look like across trials. Is all of Figure 2 one example neuron or are they different neurons? Error bars for PETH would be useful to show the reliability and robustness of the tuning.

Figure 2 shows different neurons, one from MIo and one from SIo for each task. There is shading showing ±1 standard error around the line for each direction, however this was a bit difficult to see. In addition to the other changes we have made to these figures, we made the lines smaller and darkened the error bar shading to accentuate this. We also added raster plots corresponding to the same neurons represented in Figure 2 as a supplement.

(3) Since there are only two data points, I am not sure I understand why the authors have bar graphs and error bars for graphs such as Figure 3B, Figure 5B, etc. How can one have an error bar and means with just 2 data points?

Those bars represent the standard error of the proportion. We have changed the y-axis label on these figures to make this clearer.

(4) Results in Figure 6 could be due to differential placement of the electrodes across the animals. How is this being accounted for?

Yes, this is a possibility which we have mentioned in the discussion. Even with careful placement there is no guarantee to capture a set of neurons with the exact same function in two subjects, as every individual is different. Rather we focus on analyses of data within the same animal. The purpose of Figure 6 is to show the difference between MIo and SIo, and between the two tasks, within the same subject. The more salient result from calculating the preferred direction is that there is a change in the distribution between control and nerve block within the same exact population. Discussions relating to the comparison between individuals are speculative and cannot be confirmed without the inclusion of many more subjects.

(5) For Figure 7, I would recommend showing the results of the Sham injection in the same figure instead of a supplement.

Thank you for the suggestion, we have added these results to the figure.

(6) I think the effects of the sensory block on the tongue kinematics are underexplored in Figure 7 and Figure 8. The authors could explore the deficits in tongue shape, and the temporal components of the trajectory.

Some of these effects on feeding have been explored in a previous paper, LaurenceChasen et al., 2022. We performed some additional analyses on changes to kinematics during drinking, including the number of licks per 10 second trial and the length of individual licks. The results of these are included below. We also calculated the difference in the speed of tongue movement during drinking, which generally decreased and exhibited an increase in variance with nerve block (f-test, p < 0.001). However, we have not included these figures in the main paper as they do not inform us about directionality.

**Author response image 6. sa4fig6:** Left halves of hemi-violins (black) are control and right halves (red) are nerve block for an individual. Horizontal black lines represent the mean and horizontal red lines the median. Results of two-tailed t-test and f-test are indicated by asterisks and crosses, respectively: *,† p < 0.05; **,†† p < 0.01; ***,††† p < 0.001.

(9) In Figures 9 and 10. Are the same neurons being recorded before and after the nerve block? It is unclear if the overall "population" properties are different, or if the properties of individual neurons are changing due to the nerve block.

Yes, the same neurons are being recorded before and after nerve block. Specifically, Figure 9B shows that the properties of many individual neurons do change due to the nerve block. Differences in the overall population response may be attributed to some of the units having reduced/no activity during the nerve block session.

Additionally, I recommend that the authors improve their introduction and provide more context to their discussion. Please elaborate on what you think are the main conceptual advances in your study, and place them in the context of the existing literature. By my count, there are 26 citations in this paper, 4 of which are self-citations - clearly, this can be improved upon.

Thank you for this suggestion. We have done an extensive rewrite of the Introduction and Discussion. We discussed the main conceptual advances in our study and place them in the context of the existing literature.